# CONCEPTUAL BELIEF-INFORMED REINFORCEMENT LEARNING

## ABSTRACT

Reinforcement learning (RL) has achieved significant success but is hindered by inefficiency and instability, relying on large amounts of trial-and-error data and failing to efficiently use past experiences to guide decisions. However, humans achieve remarkably efficient learning from experience, attributed to abstracting concepts and updating associated probabilistic beliefs by integrating both uncertainty and prior knowledge, as observed by cognitive science. Inspired by this, we introduce Conceptual Belief-Informed Reinforcement Learning to emulate human intelligence (HI-RL), an efficient experience utilization paradigm that can be directly integrated into existing RL frameworks. HI-RL forms concepts by extracting high-level categories of critical environmental information and then constructs adaptive concept-associated probabilistic beliefs as experience priors to guide value or policy updates. We evaluate HI-RL by integrating it into various existing value- and policy-based algorithms (DQN, PPO, SAC, and TD3) and demonstrate consistent improvements in sample efficiency and performance across both discrete and continuous control benchmarks.

## 1 INTRODUCTION

Reinforcement Learning (RL) has achieved remarkable success in various exciting areas, including aligning and enabling efficient inference of large language models Ouyang et al. (2022); Hao et al. (2025), game playing (Mnih et al., 2015), robotics (Singh et al., 2022), autonomous driving (Kiran et al., 2021), and etc. Despite these achievements, RL remains fundamentally limited by its significant sample inefficiency compared to human learning (Chiu et al., 2023; Ye et al., 2021; Joshi et al., 2025), typically relying on vast amounts of trial-and-error interactions and often struggling to generalize to unseen or sparsely observed space (states) (Mnih et al., 2015; Lake et al., 2017). In contrast, humans can quickly learn and adapt to new spaces using only a handful of experiences, highlighting a substantial gap in data efficiency between RL and human cognition (Tenenbaum et al., 2006; Lake et al., 2015; Tenenbaum et al., 2011; Griffiths et al., 2010).

The gap in learning efficiency motivates the "Era of Experience" (Silver & Sutton, 2025), which emphasizes leveraging past interactions to accelerate learning and foster new concepts and behaviors, rather than passively processing vast amounts of data. Cognitive science highlights two mechanisms for leveraging experience that are essential to human learning efficiency (Tenenbaum et al., 2011): *conceptual abstraction* and *probabilistic priors*. Conceptual abstraction distills reusable structures such as prototypes, taxonomies, causal schemas — that enable compositional reasoning, generalization, and knowledge transfer (Tenenbaum et al., 2011; Lake et al., 2015; Rosch, 1978; Kemp & Tenenbaum, 2008). In parallel, behavioral studies show that humans aggregate past experiences into adaptive probabilistic priors (Griffiths & Tenenbaum, 2005; Peterson & Beach, 1967), integrating them with future uncertainty to guide predictions and decisions (Griffiths & Tenenbaum, 2005; Tenenbaum et al., 2006).

Various RL studies leveraged either conceptual abstraction or probabilistic priors from experience independently. However, a systematic approach to combining both experience utilization mechanisms - experience-based priors grounded in extracted conceptual formations, the very mechanism underlying humans' efficient generalization—remains underexplored (Gerstenberg & Tenenbaum, 2017). Specifically, abstraction in RL has focused on representation learning approaches such as contrastive learning and bisimulation metrics to compress or align observations into compact latent

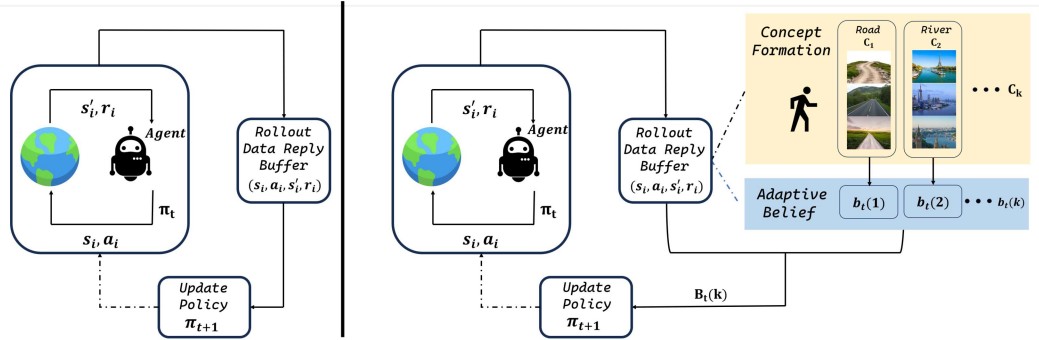

Figure 1: Traditional RL (left) replays raw transitions, while HI-RL (right) organizes them into **conceptual categories** with **adaptive beliefs**, enabling abstraction and belief-guided learning.

spaces to improve downstream task efficiency (Patil et al., 2024; Ferns et al., 2004; Castro, 2020; Peng et al., 2023). However, these methods typically do not further exploit the abstracted latent space to aggregate past experience, limiting its utility for improving RL learning efficiency. In parallel, Bayesian approaches, such as Thompson sampling, Bayesian model-based, and model-free algorithms Dearden et al. (1998) are widely used to address uncertainty and the exploration-exploitation tradeoff in RL (Dearden et al., 1998; Ghavamzadeh et al., 2015; Ross & Pineau, 2008; Thompson, 1933; Dearden et al., 1998). However, these methods are rarely integrated with conceptual abstraction.

To efficiently leverage experience, we introduce Conceptual Belief-Informed RL, named HI-RL (Human Intelligence-RL), a framework that combines conceptual abstraction and concept-based probabilistic prior, illustrated in Figure 1. HI-RL provides an algorithm-agnostic interface that integrates seamlessly with existing RL frameworks, accelerating learning by leveraging experience efficiently. It extracts concepts from large state spaces and reformulates experience into priors grounded in these abstractions, mimicking human-like conceptualization for learning efficiency. Our main contributions are summarized as below:

- We present HI-RL , an experience-utilization framework that efficiently leverages past experiences to emulate human-like learning efficiency. HI-RL reformulates the set of past experiences into probabilistic belief priors grounded in conceptual abstractions. These concept-based priors are adaptively updated over time and incorporated as auxiliary knowledge into RL value or policy updates.

- HI-RL is algorithm-agnostic and functions as a flexible module that can be seamlessly integrated into existing RL frameworks. To demonstrate its versatility, we integrate HI-RL into several popular RL algorithms (Q-learning, PPO, SAC, and TD3) and evaluate performance across both discrete and continuous tasks, achieving consistent improvements in learning efficiency and overall performance.

## 2 RELATED WORKS

### 2.1 COGNITIVE SCIENCE FOR CONCEPTUAL LEARNING

Humans achieve remarkable learning efficiency by generalizing from limited experience through Bayesian inference, integrating prior knowledge with new evidence under uncertainty (Tenenbaum & Griffiths, 2001; Griffiths & Tenenbaum, 2005; Tenenbaum et al., 2006). This supports conceptual abstraction—extracting high-level structure from sparse data—and enables causal reasoning and cross-domain transfer (Tenenbaum et al., 2011; Kemp & Tenenbaum, 2008). Recent work formalizes how learners reorganize internal knowledge via probabilistic reasoning (Lake et al., 2015; 2017), motivating the integration of such principles into machine learning for scalability, adaptability, and sample efficiency (Ma et al., 2022). Studies further show that uncovering latent causal structures enhances interpretability and abstraction, even in complex domains such as joint behavior

analysis (Gu et al., 2025; 2024). Yet, despite these advances, reinforcement learning remains dominated by replay, metric-based similarity, or policy integration, with little use of structured conceptual abstraction from cognitive science.

## 2.2 EXPERIENCE-INFORMED REINFORCEMENT LEARNING

Experience has long been exploited to improve efficiency in RL. Habit-based RL models long-term regularities as habitual priors that accelerate action selection but lack flexibility for abstraction and transfer (Daw et al., 2005; Collins & Cockburn; Keramati et al., 2011). Replay-based techniques such as PER (Schaul et al., 2015), HER and its prioritized variants (Andrychowicz et al., 2017; Sun et al., 2025; Kim et al., 2025) enhance sample efficiency by weighting or relabeling transitions, while refinements like FoDA (Chen et al., 2024) and EDER (Zhao et al., 2024) adapt distributions or promote diversity to improve generalization. Beyond replay, episodic memory models (NEC) (Pritzel et al., 2017) enable rapid value retrieval, and hybrid gradients (Q-Prop, IPG) (Gu et al., 2016; 2017) fuse on- and off-policy signals for variance reduction. Collectively, these methods leverage past interactions via sampling or memory mechanisms, yet remain confined to buffer-level operations and lack pathways for higher-order conceptual abstraction and belief-structured generalization.

## 2.3 ABSTRACTION IN REINFORCEMENT LEARNING

State abstraction has long been studied as a means to compress state spaces and enable generalization in RL (Bertsekas et al., 1988; Givan et al., 2003; Ravindran, 2004; 2003; Li et al., 2006; Kulkarni et al., 2016). Classical bisimulation and Kantorovich metrics provide strong theoretical guarantees but are computationally expensive and highly sensitive to perturbations (Ferns et al., 2004; 2011). Task-specific metrics improve offline evaluation (Pavse & Hanna, 2023) but lack adaptability, while scalable relaxations (Castro, 2020) trade rigor for tractability. Trajectory-chain and pseudometric methods (Girgin et al., 2007; Dadashi et al., 2021) offer finer granularity but incur high storage or auxiliary costs. More recent work, such as Patil et al. (2024), leverages contrastive objectives and modern Hopfield networks to compress large state spaces into abstract nodes, thereby facilitating downstream RL. These approaches primarily focus on constructing a new, compressed state space or representation for downstream algorithms. In contrast, our framework preserves the original state and exploration space while introducing an abstraction-based belief layer on top. We focus on utilizing conceptual abstraction as a basis to update its probabilistic priors, efficiently aggregating and using past experience to improve toward human-like efficient learning.

## 3 PROBLEM FORMULATION

**Markov Decision Process (MDP)** Considering reinforcement learing problems formalized as MDP (Bellman, 1957; Sutton & Barto, 2018) $\mathcal{M} = (\mathcal{S}, \mathcal{A}, \mathcal{T}, r, \mu_0, \gamma, T)$. Here $T$ is the horizon length. $\mathcal{S}$ denotes states space ($s \in S$) and $\mathcal{A}$ denotes the action spaces ($a \in A$). $\mathcal{T}(s_{t+1} \mid s_t, a_t)$ represents the transition dynamics, specifying the probability distribution over the next state $s_{t+1}$ conditioned on the current state $s_t \in \mathcal{S}$ and action $a_t \in \mathcal{A}$ at $t_{th}$ time step ($1 \le t \le T$). $r(s, a)$ represents the reward function given the state s and action a. The initial state follows $\mu_0$, $\gamma \in (0, 1)$ is the discount factor. The goal of this MDP problem is to identify an optimal policy $\pi$ that achieves the maximum expected discounted return:

$$\max_{\pi} \mathbb{E}_{\pi, \mathcal{T}, \mu_0} \Big[ \sum_{t=0}^{T} \gamma^t r(s_t, a_t) \Big]. \tag{1}$$

Formally, given the horizon length $T$ and transition dynamics $\mathcal{T}$, the long-term return from time step $t = 0$ to $t = T$ associated with the optimal policy $\pi$ is quantified through the Q-function and the Value-function (Watkins et al., 1992) by expected cumulative rewards from initial state $\mu_0$, defined as:

$$Q^\pi(s, a) = \mathbb{E}_{\pi, \mathcal{T}, \mu_0} \Big[ \sum_{t=0}^{T} \gamma^t r(s_t, a_t) \mid s_0 = s, a_0 = a \Big], V^\pi(s) = \mathbb{E}_{\pi, \mathcal{T}, \mu_0} \Big[ \sum_{t=0}^{T} \gamma^t r(s_t, a_t) \mid s_0 = s \Big]$$

$$\tag{2}$$

# 4 CONCEPTUAL BELIEF-INFORMED REINFORCEMENT LEARNING

In this section, we present Conceptual Belief-Informed Reinforcement Learning (HI-RL ), enhancing experience to emulate human intelligence learning efficiency. HI-RL consists of two core modules: (i) *Concept Formation*, which clusters state–based experiences into semantically coherent categories, and (ii) a *belief* representing probabilistic prior grounded on different concepts, defining probabilistic action experience prior over these categories. By coupling conceptual abstraction with belief-guided reasoning, HI-RL provides a structured and uncertainty-aware foundation for policy learning, supporting stable updates, efficient generalization, and reuse of past experiences.

## 4.1 CONCEPT FORMATION

The foundation for abstracting concepts can vary, as long as it represents the current situation and critical information about the environment and the agent. In this work, HI-RL focuses on state spaces, as states encapsulate essential information for decision-making and directly influence the agent's behavior and learning process, enabling pattern recognition and generalization. Specifically, we partition the states into disjoint subsets, with each subset representing a distinct concept formed by grouping states with shared characteristics and properties, thereby facilitating effective knowledge transfer within each concept. In the following, we mathematically define a conceptual abstraction as:

**Definition 4.1** (Concept Formation in State Space). A concept formation in the state space is defined as a collection of subsets $C_K = \{C_1, \ldots, C_K\}$ that satisfy $\mathcal{S} = \bigcup_{k=1}^{K} C_k$, meaning the subsets are disjoint and collectively cover the entire state space. Here, $K$ denotes a finite, prescribed number of concept categories.

In practice, conceptual abstractions can be obtained with various clustering methods. In this work, we adopt K-means (Lloyd, 1982) for its simplicity and scalability, though alternatives (e.g., spectral or hierarchical clustering) are equally applicable.

## 4.2 CONCEPTUAL ADAPTIVE BELIEF FOR RL

With abstract concepts in mind, where each concept groups states that share similar features and actions, we aggregate observed information within a concept into a unified container, a *concept-based belief*. Philosophically, a *belief* is an internal representation of how an agent interprets and anticipates the world, serving as a guide for inference and decision-making under uncertainty rather than as absolute truth (Dennett, 1988). In this work, each concept $C_k$ is paired with a time-adaptive belief $b_t(\cdot \mid k) \in \Delta(\mathcal{A})$, derived from the accumulation of past decisions and outcomes within that concept. Formally, for a conceptual abstraction $C_K = \{C_1, \ldots, C_K\}$, we define the mapping $b_t : [K] \to \Delta(\mathcal{A})$, where $b_t(\cdot \mid k)$ encodes the integrated action preferences of all states belonging to $C_k$.

We leverage the aggregated experience within each concept to accelerate learning by using concept-based beliefs as priors in RL updates. These beliefs can be seamlessly integrated into any existing RL algorithm. At each timestep $t$, we combine two signals: (i) instant feedback $\mathcal{Z}_t : \mathcal{S} \to \mathcal{A}$, defined by the base algorithm (e.g., Q-values in DQN, Gaussian policy in SAC, clipped surrogate in PPO, or deterministic actor in TD3), and (ii) the prior $b_t$, aggregated from past experience within the corresponding concept. For a given state $s \in \mathcal{S}$, we first identify its concept index $c(s)$ such that $s \in C_{c(s)}$, and then fuse the signals as:

$$B_t(\cdot \mid s) = (1 - \beta_t)\mathcal{Z}_t(\cdot \mid s) + \beta_t b_t(\cdot \mid c(s)), \tag{3}$$

where $\beta_t \in [0, 1]$ is an adaptive parameter monotonic in $t$, satisfying $\lim_{t \to \infty} \beta_t = \beta^*$ with $\beta^* \in [0, 1]$ a constant denoting the limiting weight on conceptual priors. In this formulation, $b_t(\cdot \mid c(s))$ is the empirical concept-based prior aggregated from experience, while $B_t(\cdot \mid s)$ is the fused distribution actually used for decision-making by combining $b_t$ with the instant feedback $\mathcal{Z}_t$.

This formulation ensures that the decision-making solutions for every state $s_t$ are influenced by both immediate feedback from the environment and the prior experience derived from the conceptual abstraction $C_{c(s)}$ to which it belongs.

---
**Algorithm 1** Conceptual Belief-Informed RL (HI-RL)

---
1: Initialize concept priors $b(\cdot \mid c(s))$
2: **for** $t = 1, 2, \ldots$ **do**
3:     Observe $s_t$; form $\mathcal{Z}_t(\cdot \mid s_t)$; fuse $B_t(\cdot \mid s_t) = (1 - \beta_t)\mathcal{Z}_t(\cdot \mid s_t) + \beta_t b_t(\cdot \mid c(s_t))$
4:     Sample $a_t \sim B_t$; step env $(r_t, s_{t+1})$
5:     Update policy with $B_t$ and prior experience $b_t$
6: **end for**

---

## 5   Algorithm Implementation

We extend HI-RL framework into multiple RL paradigms by developing HI-Q, HI-PPO and HI-SAC (For HI-TD3, see Appendix A.1).

### 5.1   Conceptual Belief-Informed Q-learning (HI-Q)

The classical Deep Q-learning (DQN) algorithm (Mnih et al., 2015) relies on updating the Q-function network using the greedy Bellman operator. Namely, in any iteration $t$, with the sampled batch $D_t$ and any sample $(s_i, a_i, r_i, s_i') \in D_t$ within it, the learning target of the Q-network $Q_{\theta_{t+1}}(s_i, a_i)$ would be

$$r_i + \gamma \max_{a \in \mathcal{A}} Q_{\theta_t}(s_i', a_i), \tag{4}$$

where $\theta_t$ represent the Q-network parameter at time $t$. With DQN in mind, we propose HI-Q to replace the learning target to a new one combining both the current Q-network information $Q_{\theta_t}$ and the conceptual abstraction experience prior $b_t$.

Specifically, we first introduce the construction of the concept-based belief prior $b_t(\cdot \mid k)$ at each time step $t$. Here, $b_t(\cdot \mid k)$ will be defined as the action visiting frequency summarzied over all state within the concept set $C_k$. For the discrete finite action space $\mathcal{A}$, we denote the number of visiting time over each state-action pair at time step t as $N_t(s, a)$. Then the experience prior $b_t$ of any $k$-th concept will be constructed as

$$\forall (a, k) \in \mathcal{A} \times [K]: \quad b_t(a \mid k) = \frac{\sum_{s \in C_k} N_t(s, a)}{\sum_{a' \in \mathcal{A}} \sum_{s \in C_k} N_t(s, a')}. \tag{5}$$

The update of $b_t$ is typically computational easily, since upon executing an sample tuple $(s_i, a_i, r_i, s_i')$, only the $(s_i, a_i)$-associated concept $b_t(a_i \mid c(s_i))$ will be updated.

Therefore, the combined information for any sample tuple $(s_i, a_i, r_i, s_i')$ associated with state $s_i'$ at time $t$ is defined as

$$B_t(\cdot \mid s_i') = (1 - \beta_t)q_t(\cdot \mid s_i') + \beta_t b_t(\cdot \mid c(s_i')), \tag{6}$$

where $\beta_t$ is a dynamic coefficient and $q_t(\cdot \mid s_i')$ denotes the *task-driven action-preference distribution*, typically instantiated as a smoothing distribution over $Q$-values (e.g., softmax with temperature $\tau_t$ or clipped-max with exploration mass $\delta_t$) that gradually concentrates on the greedy action as $t$ increases (Barber, 2023), computed via a softmax over the top-$k$ Q-values of state $s_i'$, effectively assigning higher probabilities to the most promising actions:

$$q_t(a \mid s_i') = \frac{\exp(Q(s_i', a)/\tau)}{\sum_{a' \in \text{top-}k(s_i')} \exp(Q(s_i', a')/\tau)}, \quad a \in \text{top-}k(s_i') \tag{7}$$

where $\tau$ is softmax temperature constant. With the constructed concept-based belief based template $B_t$ in hand, we replace the (greedy) maximum operator in Eq. 4 of classical Q-learning to a smoothed surrogate one combining both the smoothed-greedy operator of the current Q-function and the conceptual-based belief. Therefore, the new target in HI-Q for the Q-network to learn is defined as

$$r_i + \gamma \sum_{a \in \mathcal{A}} B_t(a \mid s_i')Q_t(s_i', a). \tag{8}$$

The entire algorithm is specified in Appendix A.2.1. Our conceptual-abstraction belief enables HI-Q to leverage both immediate task feedback and accumulated conceptual structures, facilitating faster learning by borrowing experience from other similar concepts.

## 5.2 CONCEPTUAL BELIEF-INFORMED PROXIMAL POLICY OPTIMIZATION (HI-PPO)

The standard PPO (Schulman et al., 2017) is a policy gradient algorithm which updates the policy by performing stochastic gradient ascent on a surrogate objective function. For any time step $t$, let $D_t$ be a sampled batch and $(s_i, a_i, r_i, s_i') \in D_t$ any individual sample within it. The objective is to update the policy $\pi_\theta(a \mid s)$ via the following loss function:

$$\mathcal{L}_{\text{PPO}} = \mathbb{E}_{(s_i, a_i) \sim \pi_{\theta_{\text{old}}}} \left[ \min \left( \frac{\pi_\theta(a_i|s_i)}{\pi_{\theta_{\text{old}}}(a_i|s_i)} A_t, \ \text{clip}\left( \frac{\pi_\theta(a_i|s_i)}{\pi_{\theta_{\text{old}}}(a_i|s_i)}, 1 - \epsilon, 1 + \epsilon \right) A_t \right) \right]. \tag{9}$$

where $\theta$ denotes the policy parameters, $A_t$ is the advantage estimate at time step $t$, and $\epsilon$ controls the trust region. While PPO updates the policy via an advantage-weighted likelihood ratio within this trust region, it depends only on immediate feedback, limiting its ability to exploit structural regularities. To overcome this, HI-PPO integrates the current policy $\pi_\theta(a \mid s)$ with the conceptual abstraction prior $b_t$.

In this paper, we focus on applying PPO in discrete action-space environments, with modifications analogous to those in HI-Q. At each time step $t$, we compute a concept-based belief prior $b_t(\cdot \mid k)$, defined as the action visitation frequency aggregated over all states in concept set $C_k$. Its computation and update follow Eq. 5, and it is combined with the policy $\pi_\theta(\cdot \mid s_i)$ for state $s_i \in D_t$ at time $t$ as:

$$B_t(\cdot \mid s_i) = (1 - \beta_t)\pi_\theta(\cdot \mid s_i) + \beta_t b_t(\cdot \mid c_k(s_i)), \tag{10}$$

where the scheduling parameter $\beta_t \in [0, 1]$ controls the influence of concept priors and increases gradually throughout training. The clipped surrogate objective of HI-PPO is:

$$\mathcal{L}_{\text{HI-PPO}} = \mathbb{E}_{(s_i, a_i) \sim \pi_{\theta_{\text{old}}}} \left[ \min \left( \frac{B_t(a_i|s_i)}{\pi_{\theta_{\text{old}}}(a_i|s_i)} A_t, \ \text{clip}\left( \frac{B_t(a_i|s_i)}{\pi_{\theta_{\text{old}}}(a_i|s_i)}, 1 - \epsilon, 1 + \epsilon \right) A_t \right) \right]. \tag{11}$$

The critic and entropy terms follow the original PPO formulation; gradients are propagated through $B_t$, allowing concept priors to steer policy updates while the clip operator guarantees trust-region stability. More implementation details and pseudocode are provided in Appendix A.2.3.

## 5.3 CONCEPTUAL BELIEF-INFORMED SOFT ACTOR-CRITIC (HI-SAC)

Traditional Soft Actor-Critic (SAC) is a maximum entropy reinforcement learning algorithm that integrates both an actor and a critic network (Haarnoja et al., 2018). Given a sampled batch $D_t$ at time step $t$, containing tuples $(s_i, a_i, r_i, s_i')$, the updates of the actor and critic networks parameters $\theta$, $\phi$ from $\pi_\theta$ and $Q_\phi$ respectively are defined as follows:

$$\mathcal{L}_{\text{critic}}(\phi_i) = \mathbb{E}\left[ \left( Q_{\phi_i}(s_i, a_i) - y_t \right)^2 \right], \quad i = 1, 2,$$
$$\text{where} \quad y_t = r_i + \gamma \, \mathbb{E}_{a_i' \sim \pi_\theta}\left[ Q_{\min}(s_i', a_i') - \alpha \log \pi_\theta(a_i' \mid s_i') \right], \tag{12}$$
$$\mathcal{L}_{\text{actor}}(\theta) = \mathbb{E}_{s_i \sim \mathcal{D}_t, \, a_i \sim \pi_\theta}\left[ \alpha \log \pi_\theta(a_i \mid s_i) - \min\{Q_{\phi_1}(s_i, a_i), Q_{\phi_2}(s_i, a_i)\} \right].$$

where $\alpha$ is entropy temperature coefficient, $y_t$ is the TD target, computed by the next state $s_i'$ at time step $t + 1$ and the corresponding action $a_i'$ sampled from the policy $\pi_\theta$. In SAC, the Q value is computed as the minimum of the estimates from two critic networks $Q_{\phi_i}$ and the actor network produces a Gaussian policy in this paper:

$$\pi_\theta(\cdot \mid s) = \mathcal{N}(\mu_{\pi_\theta}(s), \sigma_{\pi_\theta}^2(s)), \tag{13}$$

where $\mu_{\pi_\theta}(s)$ and $\sigma_{\pi_\theta}^2(s)$ denote the mean and variance predicted by the policy network for state $s$. To support concept-informed decision-making in continuous action spaces, we propose HIS-AC to integrate current actor network $\pi_{\theta_t}$ and the conceptual experience prior $b_t$. Unlike HI-Q and HI-PPO, the concept-based belief prior $b_t(k)$ constructed at each time step t is defined over the actor network parameters corresponding to the states s within the concept set $C_k$:

$$\forall (\mu, \sigma^2, k) \in \{\mu, \sigma^2\} \times [K]: \quad b_t(k) = \{\mu_{\pi_\theta}(s), \sigma_{\pi_\theta}^2(s)\}, \quad s \in C_k \tag{14}$$

During training, we update $b_t$ using a Bayesian posterior update. Let $\mu_c$ and $\sigma_c^2$ be the parameters of the current policy $\pi_{\theta_t}(s)$ and $\mu_e$ and $\sigma_e^2$ be the experience stored in $b_{t-1}(k)$:

$$b_t(k) = \left\{ \frac{\sigma_c^2 \mu_e + \sigma_e^2 \mu_c}{\sigma_c^2 + \sigma_e^2}, \ \frac{1}{\frac{1}{\sigma_c^2} + \frac{1}{\sigma_e^2}} \right\}, \quad (\mu_c, \sigma_c^2) \sim \pi_{\theta_t}(s), \quad (\mu_e, \sigma_e^2) \sim b_{t-1}(k), \quad s \in C_k \tag{15}$$

At the same time, for any sample tuple $(s_i, a_i, r_i, s_i')$ at time step $t$, we use both $s_i$ and $s_i'$ to obtain the corresponding $(\mu_{\pi_{\theta_t}}(s_i), \sigma^2_{\pi_{\theta_t}}(s_i))$ and $(\mu_{\pi_{\theta_t}}(s_i'), \sigma^2_{\pi_{\theta_t}}(s_i'))$ for the actor and critic, respectively. This fusion method can then be formally defined as:

$$\mu_{\text{actor}}(s_i) = (1 - \beta_t)\mu_{\pi_{\theta_t}}(s_i) + \beta_t \mu_b, \quad \sigma^2_{\text{actor}}(s_i) = (1 - \beta_t)\sigma^2_{\pi_{\theta_t}}(s_i) + \beta_t \sigma^2_b,$$

$$\mu_{\text{critic}}(s_i') = (1 - \beta_t)\mu_{\pi_{\theta_t}}(s_i') + \beta_t \mu_b, \quad \sigma^2_{\text{critic}}(s_i') = (1 - \beta_t)\sigma^2_{\pi_{\theta_t}}(s_i') + \beta_t \sigma^2_b, \quad (16)$$

$$\text{where} \quad (\mu_b, \sigma^2_b) \sim b_t(k), \quad s_i, s_i' \in C_k$$

where $\beta_t \in [0, 1]$ adaptively controls the relative weighting between task-driven and concept-informed signals, and $\mu_b, \sigma^2_b$ denote the currently stored conceptual experience in $b_t(k)$. This results in the conceptual belief-informed distribution for both the actor and critic:

$$B_t(\cdot \mid s_i = \mathcal{N}(\mu_{\text{actor}}(s_i), \sigma^2_{\text{actor}}(s_i)), \quad B_t(\cdot \mid s_i') = \mathcal{N}(\mu_{\text{critic}}(s_i'), \sigma^2_{\text{critic}}(s_i')). \quad (17)$$

Finally, we replace the policy $\pi_\theta$ in Eq.12 with the integrated $B_t$ and perform the updates accordingly:

$$\mathcal{L}_{\text{HI-SAC}_{critic}}(\phi_i) = \mathbb{E}\left[\left(Q_{\phi_i}(s_i, a_i) - y_t\right)^2\right], \quad i = 1, 2,$$

$$\text{where} \quad y_t = r_i + \gamma \, \mathbb{E}_{a_i' \sim B_t}\left[Q_{\min}(s_i', a_i') - \alpha \log B_t(a_i' \mid s_i')\right], \quad (18)$$

$$\mathcal{L}_{\text{HI-SAC}_{actor}}(\theta) = \mathbb{E}_{s_i \sim \mathcal{D}_t, a_i \sim B_t}\left[\alpha \log B_t(a_i \mid s_i) - \min\{Q_{\phi_1}(s_i, a_i), Q_{\phi_2}(s_i, a_i)\}\right].$$

By integrating policy learning with semantically grounded beliefs, HI-SAC enables agents to generalize across conceptually coherent behaviors. This fusion facilitates better sample reuse, long-term coherence, and more human-like decision-making. The pseudocodes are provided in Appendix A.2.2.

# 6 EXPERIMENT

**Experimental setup:** Evaluation is based on *Feasible Cumulative Rewards*, where higher values indicate better performance, averaged over three seeds (123, 321, 666). The evaluation spans a wide range of environments, including Classic Control, Box2D (Catto, 2005), MetaDrive (Li et al., 2022), MuJoCo (Todorov et al., 2012), and Atari (Bellemare et al., 2013) domains. Conceptual clustering is simulated using clustering algorithms that group similar state-action pairs into latent categories. All methods employ identical hyperparameters and are implemented on the XuanCe benchmark suite (Liu et al., 2023).

**Evaluated methods:** For discrete action spaces, we compare HI-Q and HI-PPO with the following baselines: DQN (Mnih et al., 2013), DDQN (Van Hasselt et al., 2016), DuelDQN (Wang et al., 2016), and PPO(Schulman et al., 2017), covering standard Q-value approximations, decoupled action evaluation, state-action advantage estimation, and clipped policy optimization. For continuous action spaces, HI-SAC is compared with A2C (Mnih, 2016), PPO, SAC (Haarnoja et al., 2018), and DDPG (Lillicrap, 2015), representing common policy-gradient and actor-critic methods with entropy regularization or deterministic gradients.

## 6.1 COMPARATIVE PERFORMANCE OF HI-RL AND BASELINES

To rigorously evaluate the HI-RL framework, we report results across a broad set of benchmark environments spanning both discrete and continuous action spaces (Table 1, Table 2). The tasks range from low-dimensional control (Classic Control, Box2D) to high-dimensional, perceptually rich domains (MetaDrive, MuJoCo), enabling a systematic assessment of generalization and sample efficiency under varying levels of complexity.

**Discrete Action Space:** As shown in Table 1, HI-DQN (HI-Q) consistently outperforms baselines (DQN, DDQN, Dueling DQN, PPO) across diverse discrete-action tasks. In simple settings such as *CartPole*, HI-DQN nearly reaches the performance ceiling with lower variance. In more complex tasks like *Box2D-CarRacing* and MetaDrive, HI-DQN achieves the highest rewards across all sub-tasks, demonstrating robustness and adaptability. Even in intermediate (*TOrSX*) and highly challenging scenarios (*XTOC*), HI-DQN maintains clear advantages, highlighting the effectiveness of belief-guided abstraction for stable learning under increasing complexity.

Table 1: Average cumulative rewards of HI-RL variants and baselines across discrete and continuous action environments.

| HI-RL for DQN Variants | | | | | |
| --- | --- | --- | --- | --- | --- |
| Environment/Method | HI-DQN | PPO | DQN | Duel_DQN | DDQN |
| Classic Control - CartPole | **499.78 ± 0.22** | 499.17 ± 0.83 | 478.44 ± 21.56 | 440.69 ± 59.31 | 396.51 ± 103.49 |
| Classic Control - Acrobot | **-80.57 ± 17.48** | -500.00 ± 0.00 | -87.19 ± 18.55 | -104.53 ± 54.19 | -100.77 ± 24.79 |
| Box2d - CarRacing | **854.66 ± 45.35** | 189.05 ± 56.48 | 830.78 ± 51.61 | -13.05 ± 24.66 | 766.16 ± 88.22 |
| Box2d - LunarLander | **232.73 ± 40.20** | 204.95 ± 48.77 | 52.67 ± 192.08 | -58.97 ± 4.08 | 191.79 ± 69.16 |
| MetaDrive - rXTSC | **189.22 ± 63.71** | 156.74 ± 31.44 | 82.05 ± 82.84 | 39.50 ± 7.27 | 185.55 ± 107.80 |
| MetaDrive - TOrSX | **159.39 ± 38.40** | 149.97 ± 26.28 | 101.60 ± 13.72 | 69.16 ± 14.07 | 83.77 ± 22.37 |
| MetaDrive - XTOC | **303.15 ± 50.89** | 293.72 ± 66.42 | 170.73 ± 31.60 | 67.42 ± 6.29 | 170.73 ± 31.60 |
| MetaDrive - XTSC | **233.91 ± 64.92** | 191.50 ± 39.31 | 215.94 ± 205.74 | 63.47 ± 4.96 | 147.71 ± 92.55 |
| MetaDrive - CYrXT | **97.99 ± 25.43** | 97.83 ± -38.66 | 77.23 ± 47.94 | 9.12 ± 39.53 | 75.39 ± 49.99 |
| MetaDrive - COrXSrT | **117.90 ± 24.56** | 89.27 ± 23.52 | 117.18 ± 30.28 | 53.01 ± 4.91 | 29.15 ± 16.26 |
| MetaDrive - SrOYCtryS | **130.27 ± 117.07** | 75.38 ± 8.12 | 105.01 ± 88.37 | 38.90 ± 0.39 | 100.72 ± 81.92 |
| HI-RL for SAC Variants | | | | | |
| Environment/Method | HI-SAC | SAC | PPO | DDPG | A2C |
| Box2d - BipedalWalker | **295.16 ± 99.64** | 285.71 ± 11.43 | -17.21 ± 45.45 | -34.58 ± 8.92 | -115.66 ± 1.95 |
| Mujoco - Ant | **2862.15 ± 606.91** | 2386.54 ± 489.76 | 108.47 ± 14.97 | 2351.56 ± 147.15 | 1566.19 ± 346.25 |
| Mujoco - Humanoid | **3248.46 ± 812.84** | 2090.07 ± 2233.68 | 52.35 ± 0.08 | 401.39 ± 84.60 | 179.26 ± 74.62 |
| Mujoco - HumanoidStandup | **132391.49 ± 606.23** | 121643.72 ± 25.53 | 112603.41 ± 65.06 | 69209.17 ± 14951.33 | 80250.37 ± 46.46 |
| Mujoco - Reacher | **-3.96 ± 0.71** | -4.65 ± 1.77 | -6.88 ± 0.08 | -5.73 ± 0.96 | -10.88 ± 0.12 |
| Mujoco - HalfCheetah | **10276.66 ± 2448.76** | 9678.01 ± 810.58 | 7378.66 ± 1951.02 | 3574.82 ± 2267.63 | 3043.32 ± 388.69 |
| Mujoco - Hopper | **3121.56 ± 573.84** | 2246.74 ± 657.82 | 1530.17 ± 1869.52 | 2338.46 ± 1075.83 | 520.53 ± 25.98 |
| Mujoco - Walker2d | **4444.48 ± 292.20** | 3382.66 ± 1177.36 | 992.81 ± 1799.20 | 3756.60 ± 840.68 | 733.50 ± 755.30 |
| Mujoco - Pusher | **-25.44 ± 6.16** | -31.76 ± 4.15 | -36.36 ± 0.82 | -45.50 ± 3.14 | -55.29 ± 1.65 |
| Mujoco - InvertedPendulum | **998.13 ± 1.87** | 860.78 ± 590.78 | 609.51 ± 4.51 | 973.82 ± 26.18 | 991.25 ± 116.64 |
| Mujoco - InvertedDoublePendulum | **9247.71 ± 103.30** | 8703.18 ± 644.18 | 126.87 ± 56.87 | 6444.11 ± 3857.15 | 7981.28 ± 1365.03 |

Table 2: Average cumulative rewards of HI-PPO, HI-TD3 and baselines across discrete and continuous action environments.

| HI-RL for PPO Variants | | | | | |
| --- | --- | --- | --- | --- | --- |
| Method/Environment | Atari - AirRaid | Atari - Amidar | Atari - Asteroids | Atari - Centipede | Atari - Zaxxon |
| PPO | 7210.01 ± 1594.32 | 917.56± 65.08 | 4190.79± 928.38 | 4792.76± 1244.33 | 15690.27± 3486.71 |
| HI-PPO | **9659.79 ± 2333.36** | **2302.55 ± 627.01** | **4419.23± 1404.85** | **6002.09± 1495.15** | **16663.71±5093.18** |

| HI-RL for TD3 Variants | | | | | |
| --- | --- | --- | --- | --- | --- |
| Method/Environment | Box2d - BipedalWalker | Mujoco - Ant | Mujoco - Swimmer | Mujoco - HalfCheetah | Mujoco - Walker2d |
| TD3 | 276.03 ± 42.42 | 5634.15± 620.63 | 50.50± 1.45 | 13194.89± 755.84 | 4565.46± 147.63 |
| HI-TD3 | **291.86 ± 23.45** | **6358.90± 420.75** | **132.89± 1.99** | **13706.98±399.64** | **6194.91± 319.83** |

**Continuous Action Space:** A similar trend is observed in continuous-control benchmarks (Table 1). HI-SAC consistently outperforms SAC, PPO, and DDPG across both medium- and high-dimensional MuJoCo and Box2D tasks. In challenging domains such as *Humanoid* and *Humanoid-Standup*, HI-SAC achieves substantially higher rewards with improved stability, while in locomotion tasks (*HalfCheetah*, *Walker2d*), it converges faster and produces more resilient policies. Overall, these results demonstrate that HI-SAC leverages belief-guided generalization to deliver reliable gains in environments requiring both precise control and long-horizon reasoning.

## 6.2 LEARNING DYNAMICS WITH EXPERIENCE-DRIVEN ABSTRACTION

While the previous section demonstrates that HI-RL achieves superior final performance over baseline algorithms in both discrete and continuous action spaces, practical reinforcement learning often places greater emphasis on sample efficiency, training stability, and convergence speed than on post-convergence metrics. These factors are especially critical in resource-constrained or high-risk settings. To this end, we analyze the learning dynamics of HI-PPO vs. PPO and HI-TD3 vs. TD3 on Atari and MuJoCo (Fig. 2, Table 2), illustrating how HI-RL leverages cognitive belief priors for faster exploration and structured abstraction for more stable optimization.

In high-dimensional visual environments such as Atari, HI-PPO consistently improves both convergence speed and final performance. For example, in *Amidar*, HI-PPO surpasses 2000 reward at 40M steps, whereas PPO converges around $\sim 900$. In more challenging tasks such as *Asteroids* and *Centipede*, HI-PPO not only learns faster but also exhibits reduced variance, indicating more stable policy updates. The progressive increase of $\beta_t$ enables HI-PPO to exploit conceptual priors early on and transition smoothly to task-specific fine-tuning, resulting in efficient and robust learning.

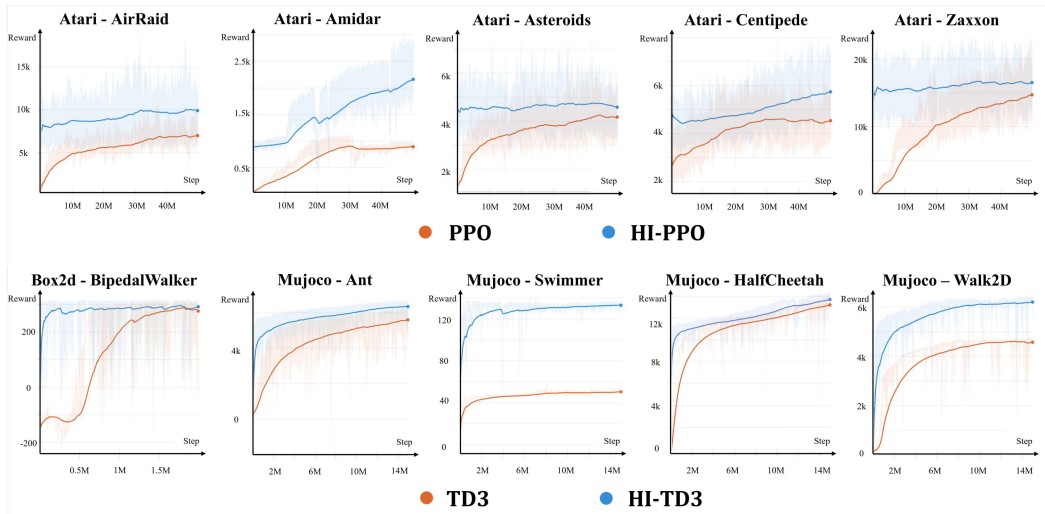

Figure 2: Learning curves comparing HI-PPO and PPO (Atari tasks) as well as HI-TD3 and TD3 (Mujoco and Box2D tasks). HI-RL variants demonstrate faster convergence, higher sample efficiency, and reduced variance across diverse environments.

Similarly, in continuous control tasks, HI-TD3 achieves faster convergence, higher rewards, and greater stability compared to TD3. In simpler tasks such as *BipedalWalker*, HI-TD3 converges more rapidly and attains comparable or better final performance. In more complex locomotion tasks including *Ant*, *Swimmer*, *HalfCheetah*, and *Walker2d*, HI-TD3 not only reaches higher asymptotic rewards but also produces smoother learning curves with lower variance. By contrast, TD3 often suffers from slower convergence and mid-training stagnation, underscoring the efficiency and robustness advantages of HI-TD3.

# 7    CONCLUSION

We introduce Conceptual Belief-Informed Reinforcement Learning (HI-RL), a representation-level framework that organizes experiences into conceptual categories and integrates belief-guided fusion into policy learning. Moving beyond buffer replay and static policy libraries, HI-RL establishes a structured memory that supports abstraction, reuse, and generalization. Across Q-learning, PPO, TD3, and SAC, it consistently improves sample efficiency, final returns, and stability in both discrete and continuous domains. By achieving higher returns with fewer interactions and stabilizing updates, HI-RL also reduces computational cost, underscoring its potential for sustainable and resource-efficient training. More broadly, HI-RL illustrates how cognitive principles—conceptual abstraction and belief—can be operationalized to advance reinforcement learning, shifting the field from raw data manipulation toward structured, human-aligned inference. We view this as a step toward an "Era of Experience," in which intelligence is grounded in the active organization of interaction history rather than rote prediction from data.

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
