# A APPENDIX

## A.1 CONCEPTUAL BELIEF-INFORMED TWIN DELAYED DEEP DETERMINISTIC POLICY GRADIENT (HI-TD3)

TD3 (Twin Delayed Deep Deterministic Policy Gradient)(Fujimoto et al., 2018), built upon DDPG(Silver et al., 2014), mitigates Q-value overestimation and improves stability via twin Q-networks, delayed updates, and target policy smoothing. Here, we focus only on the actor update. Considering a sample batch $D_t = (s_i, a_i, r_i, s_i')$ at time step t, the actor policy update is defined as:

$$\mathbb{E}_{s_i \sim D_t} \big[ Q_{\phi_{\min}}(s_i, \pi_\theta(s_i)) \big] \tag{19}$$

where $Q_{\phi_{\min}}$ takes the smaller value of the two Q-networks, while $\pi_\theta$ denotes the policy that generates actions, with $a_i = \pi_\theta(s_i)$. Conceptual Belief-Informed TD3 (HI-TD3) applies the HI-RL fusion rule to deterministic policy gradients, refining the concept-based belief prior $b_t(k)$ for each conceptual category $C_k$ at time step t as:

$$\forall (\nabla, k) \in \nabla \times [K]: \quad b_t(k) \simeq \nabla_a Q_{\phi_{\min}}(s, a) \tag{20}$$

where $b_t(k)$, a directional belief, denotes as $\nabla_a Q_{\min}(s, a)$, where the smaller Q-value in TD3 is employed to approximate the gradient serving as its representation. The recorded belief direction is updated using an exponential moving average with normalization:

$$b_t(k) = \frac{(1-\eta)b_{t-1}(k) + \eta \nabla_a Q_{\min}(s, a)}{\|(1-\eta)b_{t-1}(k) + \eta \nabla_a Q_{\min}(s, a)\|}, \quad a \sim \pi_\theta(s), \quad s \in C_k \tag{21}$$

where $\eta$ is an exponential moving average constant and $b_{t-1}(k)$ denotes the previously stored directional belief.

In the policy update of HI-TD3, we perform belief fusion updates only on the actor network. At each time step $t$ with sampled tuple $(s_i, a_i, r_i, s_i')$, the integrated directional belief information $B_t(k)$ is denotes as:

$$B_t(k) = c \frac{(1-\beta)\nabla_{a_i} Q_{\min}(s_i, a_i) + \beta b_t(k)}{\|(1-\beta)\nabla_{a_i} Q_{\min}(s_i, a_i) + \beta b_t(k)\|}, \quad s_i \in C_k \tag{22}$$

where $c$ is a constant used to prevent excessive oscillations if $B_t(k)$ becomes too large. Differing from previous usage, $\beta$ is determined by directional similarity, computed as a dot product, and serves as the fusion coefficient:

$$\beta = \text{clamp}\Big(\sum_k b_t(k) \cdot \nabla_{a_i} Q_{\phi_{\min}}(s_i, a_i), 0, 1\Big) \tag{23}$$

The directional fusion is performed by combining $B_t(k)$ as a perturbation with $a_i$:

$$a_{\text{blend}} = \text{clamp}\big(a_i + B_t(k), -1, 1\big) \tag{24}$$

The actor minimizes to update policy:

$$\mathbb{E}_{s_i \sim D_t} \big[ Q_{\phi_{\min}}(s_i, a_{blend}) \big] \tag{25}$$

Thus, HI-TD3 preserves the HI-RL fusion principle through the actor by blending task-driven gradients with conceptual priors, while the critic remains the standard TD3 update for stability. This makes HI-TD3 a deterministic yet framework-consistent instantiation of HI-RL. The pseudocodes are provided in Appendix A.2.4.

## A.2 PSEUDO CODE

### A.2.1 CONCEPTUAL BELIEF-INFORMED Q-LEARNING (HI-Q) ALGORITHM

---

**Algorithm 2** Conceptual Belief-Informed Q-learning (HI-Q) Algorithm

---

1: Initialization: learning rate $\alpha$, discount factor $\gamma$, running steps $T$, episodes $E$, replay buffer $\mathcal{B}$ and a set of K conceptual categories, denoted as $\{\mathcal{C}_k\}_{k=1}^{K}$
2: **for** each episode **do**
3:     Get initial state $s_0$ from the environment
4:     **for** each timestep t **do**
5:         Choose a random action $a_t$ with probability $\epsilon$ otherwise take $a_t = \arg\max_a Q(s_t, a; \theta)$
6:         Execute $a_t$ to get reward $r(s_t, a_t)$, next state $s_{t+1}$
7:         Store $(s_t, a_t, r(s_t, a_t), s_{t+1})$ into $\mathcal{B}$
8:         Identify the conceptual category $\mathcal{C}_k$ of $s_t$ through Nearest Neighbor
9:         Update the count of $a_t$ in $\mathcal{C}_k$ (cf. 5);
10:        Sample $N$ tuples from $\mathcal{B}$ to update $Q$ function:
11:           Extract $b_t(a \mid \mathcal{C}_k(s_t))$ and integrate with rewards to estimate $B_t(a \mid s_{t+1})$ (cf.6)
12:           $y_{s_t,a_t}^{i} = \mathbb{E}_{\mathcal{B}}\left[r(s_t, a_t) + \gamma \sum_a B_t(a \mid s_{t+1})Q(s_{t+1}, a; \theta^-)|s_t, a_t\right]$ (cf.8)
13:           $Loss = \mathbb{E}_{\mathcal{B}}\left[(y_{s_t,a_t}^{i} - Q(s_t, a_t; \theta))^2\right]$
14:         Reset target network after a few updates: align target Q parameters: $\theta^- = \theta$;
15:     **end for**
16: **end for**

---

### A.2.2 CONCEPTUAL BELIEF-INFORMED SOFT ACTOR-CRITIC (HI-SAC) ALGORITHM

---

**Algorithm 3** Conceptual Belief-Informed Soft Actor-Critic

---

1: Initialize two critic parameters $\phi_1$, $\phi_2$ and actor parameters $\theta$, Conceptual categories $\{C_k\}_{k=1}^N$, category belief parameters $b_{t=0}(k) = \{\mu_k, \sigma_k^2\}_{k=1}^N$
2: **for** each time step t **do**
3:     Sample $a_t \sim \pi_\theta(\cdot \mid s_t)$
4:     Transition to $s_{t+1} \sim p(s_{t+1} \mid s_t, a_t)$
5:     Store transition in replay buffer: $\mathcal{B} \leftarrow \mathcal{B} \cup \{(s_t, a_t, r(s_t, a_t), s_{t+1})\}$
6:     **for** training step **do**
7:         Sampled $\{s_i, a_i, r_i, s_i'\} \leftarrow \mathcal{B}$
8:         Identify category $C_k$ for $s_i$ through Euclidean distance
9:         Compute $B_t(si)$ and $B_t(si')$, $s_i, s_i' \in C_K$ (cf. 17)
10:        Update category belief parameters $b_t(k)$ (cf. 15)
11:     **end for**
12:     **for** each gradient step (cf.18) **do**
13:         Compute target:

$$y_i = r_i + \gamma \, \mathbb{E}_{a_i' \sim B_t(\cdot|s_i')}\Big[Q_{\min}(s_i', a_i') - \alpha \log B_t(a_i'|s_i')\Big].$$

14:         Update critics $(i = 1, 2)$:

$$L_{\text{CBISAC}}^{\text{critic}}(\phi_i) = \mathbb{E}_{(s_i, a_i) \sim D_t}\Big[\big(Q_{\phi_i}(s_i, a_i) - y_i\big)^2\Big],$$

$$\phi_i \leftarrow \phi_i - \eta_\phi \nabla_{\phi_i} L_{\text{CBISAC}}^{\text{critic}}(\phi_i).$$

15:         Update actor:

$$L_{\text{CBISAC}}^{\text{actor}}(\theta) = \mathbb{E}_{s_i \sim D_t, \, a_i \sim B_t}\Big[\alpha \log B_t(a_i|s_i) - Q_{\min}(s_i, a_i)\Big],$$

$$\theta \leftarrow \theta - \eta_\theta \nabla_\theta L_{\text{CBISAC}}^{\text{actor}}(\theta).$$

16:         Update temperature:

$$L(\alpha) = \mathbb{E}_{s_i, a_i \sim B_t}\big[-\alpha\left(\log B_t(a_i|s_i) + \mathcal{H}_{\text{target}}\right)\big],$$

$$\alpha \leftarrow \alpha - \eta_\alpha \nabla_\alpha L(\alpha).$$

17:         Soft update target network:

$$\bar{\phi} \leftarrow \tau\phi + (1 - \tau)\bar{\phi}.$$

18:         Update $b_t(k)$ (cf.15)
19:     **end for**
20: **end for**

---

### A.2.3 CONCEPTUAL BELIEF-INFORMED PROXIMAL POLICY OPTIMIZATION (HIPPO) ALGORITHM

---

**Algorithm 4** Conceptual Belief-Informed Proximal Policy Optimization

---

1: Initialize policy parameters $\theta$ and value function parameters $\phi$, conceptual categories $\{C_k\}_{k=1}^{N}$
2: **for** each iteration **do**
3:     **for** each environment step t **do**
4:         Collect set of trajectories $D_k = \{\tau_i\}$ by running $\pi_k = \pi(\theta_k)$
5:         Sample $a_t$ and Transition to get $s_{t+1}$
6:         Compute rewards-to-go $r(s_t, a_t)$.
7:         Compute advantage estimation $A_t$ based on current value function $V_{\phi_k}$
8:         Store transition in replay buffer: $\mathcal{B} \leftarrow \mathcal{B} \cup \{(s_t, a_t, r(s_t, a_t), s_{t+1}, A_t)\}$
9:     **end for**
10:     **for** each gradient step **do**
11:         Sampled $\{s_i, a_i, r_i, s_i'\} \leftarrow \mathcal{B}$
12:         Identify category $C_k$ for $s_i$ through Euclidean distance
13:         Compute $B_t(k) = (1 - \beta_t)\pi_\theta(a_i \mid s_i) + \beta_t b_t(k), \quad s_i \in C_k$ (cf.10)
14:         Update the policy by maximizing the PPO-Clip objective (cf.11):
15:

$$\theta_{k+1} \;=\; \arg\max_\theta \; \mathbb{E}_{(s_i,a_i)\sim\pi_{\theta_{\text{old}}}} \left[ \min\left( \frac{B_t(a_i \mid s_i)}{\pi_{\theta_{\text{old}}}(a_i \mid s_i)} A_i, \; \text{clip}\left( \frac{B_t(a_i \mid s_i)}{\pi_{\theta_{\text{old}}}(a_i \mid s_i)}, \; 1-\epsilon, \; 1+\epsilon \right) A_i \right) \right].$$

16:         Fit value function by regression on mean-squared error:
17:

$$\phi_{k+1} \;=\; \arg\min_\phi \; \mathbb{E}_{(s_i,a_i)\sim\pi_{\theta_{\text{old}}}} \left[ \left( V_\phi(s_i) - r(s_i, a_i) \right)^2 \right].$$

18:         Update $b_t(k)$ (cf.5)
19:     **end for**
20: **end for**

---

### A.2.4 Conceptual Belief-Informed Twin Delayed Deep Deterministic (HI-TD3) Algorithm

---

**Algorithm 5** Conceptual Belief-Informed Twin Delayed Deep Deterministic

---

1: Initialize actor $\pi_\theta(s)$, critics $Q_{\phi_1}(s,a), Q_{\phi_2}(s,a)$, target networks $\theta' \leftarrow \theta, \phi'_1 \leftarrow \phi_1, \phi'_2 \leftarrow \phi_2$, Conceptual categories $\{C_k\}_{k=1}^N$, discount factor $\gamma$
2: **for** each iteration **do**
3:      **for** each environment step t **do**
4:          Sample $a_t = \pi_\theta(s_t) + \epsilon, \quad \epsilon \sim \mathcal{N}(0, \sigma^2 I)$
5:          Transition to $s_{t+1} \sim p(s_{t+1} \mid s_t, a_t)$
6:          Store transition in replay buffer: $\mathcal{B} \leftarrow \mathcal{B} \cup \{(s_t, a_t, r(s_t, a_t), s_{t+1})\}$
7:      **end for**
8:      **for** each gradient step **do**
9:          Sample minibatch $D_t = \{(s_i, a_i, r_i, s'_i)\}$ from replay buffer
10:          **Critic update (TD3)**
11:            Compute target action with smoothing: $a'_i = \pi_{\theta'}(s'_i) + \epsilon, \quad \epsilon \sim \mathcal{N}(0, \sigma^2 I)$
12:            Compute temporal difference target: $y_i = r_i + \gamma \cdot \min\left(Q_{\phi'_1}(s'_i, a'_i), Q_{\phi'_2}(s'_i, a'_i)\right)$
13:            Update critics by minimizing loss: $L = \frac{1}{N}\sum_i \left(y_i - Q_{\phi_j}(s_i, a_i)\right)^2, \quad j = 1, 2$
14:          **Actor update with fusion**
15:            Compute gradient direction: $g_t = \nabla_a Q_{\min}(s_i, a_i)$
16:            Compute fusion coefficient $\beta$ (cf.23)
17:            Integrate belief: $B_t = c \cdot \frac{(1-\beta)g_t + \beta b_t(k)}{\|(1-\beta)g_t + \beta b_t(k)\|}$ (cf.25)
18:            Blend action: $a_{\text{blend}} = \text{clamp}(\pi_\theta(s_i) + B_t, -1, 1)$(cf.24)
19:            Update actor by minimizing: $J(\theta) = -\frac{1}{N}\sum_i Q_{\min}(s_i, a_{\text{blend}})$(cf.21)
20:          **Target network updates**

$$\phi' \leftarrow \tau\phi + (1-\tau)\phi', \quad \theta' \leftarrow \tau\theta + (1-\tau)\theta'$$

21:          Update $b_t(k)$ (cf.22)
22:      **end for**
23: **end for**

---

## A.3 Smoothed Bellman Operator

To reflect cognitive properties of uncertainty-aware decision-making in reinforcement learning, we revise the classical Bellman operator, which updates values deterministically:

$$\mathcal{T}Q(s_t, a_t) = r_t + \gamma \max_{a \in \mathcal{A}} Q_t(s_{t+1}, a). \tag{26}$$

Here, $\mathcal{T}$ is the classical Bellman operator, $s_t \in \mathcal{S}$ denotes the current state, $a_t \in \mathcal{A}$ the chosen action, $r_t \in \mathbb{R}$ the immediate reward, $\gamma \in (0, 1)$ the discount factor, $\mathcal{A}$ the action space, and $Q_t(s, a)$ the action–value function at iteration $t$. The $\max_{a \in \mathcal{A}}$ term represents greedy action selection, i.e., propagating value based on the action with the highest estimated return.

$$\mathcal{T}_{\text{Smoothed}}Q(s_t, a_t) = r_t + \gamma \sum_{a \in \mathcal{A}} q_t(a \mid s_{t+1})Q_t(s_{t+1}, a), \tag{27}$$

where $\mathcal{T}_{\text{Smoothed}}$ denotes the *Smoothed Bellman Operator*, which replaces the hard maximization with a belief-weighted expectation. Here, $q_t(a \mid s_{t+1})$ is the action-preference distribution at state $s_{t+1}$, e.g., a softmax distribution over $Q_t(s_{t+1}, a)$ or the belief-preference distribution in HI-RL. Unlike the deterministic max, this formulation propagates value in a probabilistic, uncertainty-aware manner, balancing task-driven estimates with belief-informed priors.

This smoothing relaxes the deterministic backup, enabling value propagation to account for uncertainty and preference variability. The Smoothed Bellman Operator thus provides a unified, differentiable mechanism for propagating reward uncertainty. In Section 5, we illustrate how Smoothed

Bellman Operator integrates with different policy learning paradigms (Q-learning, SAC, PPO). Formal instantiations such as softmax smoothing, clipped interpolation, and Bayesian fusion are provided in next, along with a convergence proof and a Jensen-type inequality.

**Lemma A.1** (Jensen's Inequality for Q-values). *Consider an MDP with state $s_{t+1}$ and actions $a$, along with Q-value estimates $Q_t(s_{t+1}, a)$. Let $q_t(a \mid s_{t+1})$ denote the probability of selecting action $a$ in state $s_{t+1}$. By Jensen's inequality:*

$$\gamma \sum_{s_{t+1}} P(s_{t+1} \mid s_t, a_t) \sum_{a'} q_t(a \mid s_{t+1}) Q_t(s_{t+1}, a) \le$$
$$\gamma \sum_{s_{t+1}} P(s_{t+1} \mid s_t, a_t) \max_a Q_t(s_{t+1}, a), \tag{28}$$

Lemma A.1 establishes that the Smoothed Bellman Operator provides a conservative backup: replacing the hard maximization with a belief-weighted expectation yields an update that forms a lower bound on the classical Bellman backup, thereby stabilizing value propagation under uncertainty.

**Lemma A.2** (Convergence of Smoothed Bellman Operator). *Let $\{Q_t\}$ be the sequence generated by iteratively applying $\mathcal{T}_{Smoothed}$. Under the condition:*

$$\lim_{t \to \infty} \max_a q_t(a \mid s_{t+1}) = 1, \tag{29}$$

*for the optimal action, $Q_t$ converges to the optimal $Q^*$ as $t \to \infty$. See Appendix D for a detailed proof.*

Lemma A.2 complements this by showing that if the action-preference distribution $q_t(\cdot \mid s_{t+1})$ asymptotically collapses onto the optimal action, then iterative application of the Smoothed Bellman Operator converges to the optimal value function $Q^*$. Together, these results establish that the Smoothed Bellman Operator not only smooths value backups for improved robustness, but also preserves the fundamental convergence guarantees of classical reinforcement learning. The full proof of Lemma A.2 is provided in next subsection.

To instantiate the Smoothed Bellman Operator in practice, different smoothing strategies can be employed to construct the action-preference distribution $b_t$. These strategies determine how strongly the update deviates from hard maximization and how uncertainty is incorporated. Representative examples are summarized in Table 3.

| Strategy | Formula |
|---|---|
| Softmax | $q_t = \dfrac{e^{Q(s,a)}}{\sum_b e^{Q(s,b)}}$ |
| Clipped Max | $q_t = \begin{cases} 1 - \tau, & \text{if } a = a^* \\ \frac{\tau}{A-1}, & \text{if } a \ne a^* \end{cases}$ |
| Clipped Softmax | $q_t = \begin{cases} \dfrac{e^{\beta Q(s,a)}}{\sum_{b \in I} e^{\beta Q(s,b)}}, & \text{if } a \in I \\ 0, & \text{if } a \notin I \end{cases}$ |

Table 3: Smoothing strategies with respective formulas

### A.3.1 CONVERGENCE PROOF

We outline a proof that builds upon the following result (Singh et al., 2000; Barber, 2023) and follows the framework provided in (Melo, 2001):

**Theorem A.3.** *The random process $\{\Delta_t\}$ taking value in $\mathbb{R}$ and defined as*

$$\Delta_{t+1}(x) = (1 - \alpha_t(x))\Delta_t(x) + \alpha_t(x) F_t(x) \tag{30}$$

*converges to 0 with probability 1 under the following assumptions:*

- $0 \le \alpha_t \le 1$, $\sum_t \alpha_t(x) = \infty$, $\sum_t \alpha_t^2(x) < \infty$;

- $\mathbb{E}[\|F_t(x)\|_W] \leq \kappa \|\Delta_t\|_W + c_t$, $\kappa \in [0, 1)$ *and* $c_t \to 0$ *with probability 1;*

- $var(F_t(x)) \leq C(1 + \|\Delta_t\|_W)^2$, $C > 0$

*where* $\|\Delta_t\|_W$ *denotes a weighted max norm.*

We are interested in the convergence of $Q_t$ towards the optimal value $Q_*$ and therefore define

$$\Delta_t = Q_t(s_t, a_t) - Q_*(s_t, a_t) \tag{31}$$

It is convenient to write the smoothed update as

$$Q_{t+1}(s_t, a_t) = Q_t(s_t, a_t) + \alpha_t(s_t, a_t)\left(r_t + \gamma \langle Q(s_{t+1}, a)\rangle_a - Q_t(s_t, a_t)\right) \tag{32}$$

where $\langle f(x)\rangle_x$ means the expectation of the function $f(x)$ with respect to the distribution of $x$. Using the smoothed update, we can write

$$\Delta_{t+1}(s_t, a_t) = Q_{t+1}(s_t, a_t) - Q_*(s_t, a_t) \tag{33}$$

$$= (1 - \alpha_t)\Delta_t + \alpha_t\left(r_t + \gamma\langle Q(s_{t+1}, a)\rangle_a - Q_*(s_t, a_t)\right) \tag{34}$$

In terms of Theorem 1, we therefore define

$$F_t = r_t + \gamma \sum_a q_t(a|s_{t+1})Q_t(s_{t+1}, a) - Q_*(s_t, a_t) \tag{35}$$

*Proof.* For convergence, we need to verify the conditions of Theorem 1.

**Step 1: Verify Step-Size Conditions**

We assume that the learning rates $\alpha_t(s_t, a_t)$ satisfy:

- $0 < \alpha_t(s_t, a_t) \leq 1$,

- $\sum_t \alpha_t(s_t, a_t) = \infty$,

- $\sum_t \alpha_t^2(s_t, a_t) < \infty$.

An example is $\alpha_t(s_t, a_t) = \frac{1}{N_t(s_t, a_t)}$, where $N_t(s_t, a_t)$ is the visitation count of $(s_t, a_t)$.

**Step 2: Establish Boundedness of $Q_t$**

Since the rewards $r_t$ are bounded ($|r_t| \leq R_{\max}$) and the discount factor $0 < \gamma < 1$, we can show that $Q_t$ remains bounded independently of the convergence of $\Delta_t$.

*Define the Bound $Q_{max}$:*

We define

$$Q_{\max} = \frac{R_{\max}}{1 - \gamma}. \tag{36}$$

This is the maximum possible value of the Q-function given the bounded rewards and discount factor.

*Derivation of $Q_{max}$:*

The Q-function $Q(s, a)$ represents the expected cumulative discounted reward when starting from state $s$ and taking action $a$:

$$Q(s, a) = \mathbb{E}\left[\sum_{k=0}^{\infty} \gamma^k r_{t+k} \,\Big|\, s_t = s, a_t = a\right], \tag{37}$$

where $r_{t+k}$ is the reward received at time $t + k$, and $\gamma$ is the discount factor.

Assuming that at each time step, the agent receives the maximum possible reward $R_{\max}$, the maximum possible Q-value is:

$$Q_{\max} = \sum_{k=0}^{\infty} \gamma^k R_{\max} = R_{\max} \sum_{k=0}^{\infty} \gamma^k. \tag{38}$$

Since $0 < \gamma < 1$, the infinite sum $\sum_{k=0}^{\infty} \gamma^k$ is a geometric series that sums to:

$$\sum_{k=0}^{\infty} \gamma^k = \frac{1}{1 - \gamma}. \tag{39}$$

Therefore, we have:

$$Q_{\max} = R_{\max} \times \frac{1}{1 - \gamma} = \frac{R_{\max}}{1 - \gamma}. \tag{40}$$

Thus, $Q_{\max} = \frac{R_{\max}}{1 - \gamma}$ is the maximum possible value of the Q-function in any state-action pair.

*Base Case:* Let $Q_0(s, a)$ be initialized such that $|Q_0(s, a)| \leq Q_{\max}$ for all $s, a$.

*Inductive Step:* Assume $|Q_t(s, a)| \leq Q_{\max}$ for all $s, a$. We need to show that $|Q_{t+1}(s_t, a_t)| \leq Q_{\max}$. From the update equation:

$$Q_{t+1}(s_t, a_t) = Q_t(s_t, a_t) + \alpha_t(s_t, a_t) \left( r_t + \gamma \left\langle Q_t(s_{t+1}, a) \right\rangle_a - Q_t(s_t, a_t) \right). \tag{41}$$

Simplifying:

$$Q_{t+1}(s_t, a_t) = (1 - \alpha_t(s_t, a_t))Q_t(s_t, a_t) + \alpha_t(s_t, a_t) \left( r_t + \gamma \left\langle Q_t(s_{t+1}, a) \right\rangle_a \right). \tag{42}$$

Taking absolute values:

$$|Q_{t+1}(s_t, a_t)| \leq (1 - \alpha_t)|Q_t(s_t, a_t)| + \alpha_t \left( |r_t| + \gamma \left| \left\langle Q_t(s_{t+1}, a) \right\rangle_a \right| \right). \tag{43}$$

Using the inductive hypothesis and boundedness:

$$|Q_t(s_t, a_t)| \leq Q_{\max}, \quad \left| \left\langle Q_t(s_{t+1}, a) \right\rangle_a \right| \leq Q_{\max}, \tag{44}$$

and $|r_t| \leq R_{\max}$. Therefore:

$$|Q_{t+1}(s_t, a_t)| \leq (1 - \alpha_t)Q_{\max} + \alpha_t \left( R_{\max} + \gamma Q_{\max} \right). \tag{45}$$

Simplify:

$$|Q_{t+1}(s_t, a_t)| \leq Q_{\max} - \alpha_t Q_{\max} + \alpha_t \left( R_{\max} + \gamma Q_{\max} \right) \tag{46}$$
$$= Q_{\max} + \alpha_t \left( R_{\max} - (1 - \gamma)Q_{\max} \right). \tag{47}$$

Since $Q_{\max} = \frac{R_{\max}}{1 - \gamma}$, we have $(1 - \gamma)Q_{\max} = R_{\max}$. Substituting back:

$$|Q_{t+1}(s_t, a_t)| \leq Q_{\max} + \alpha_t \left( R_{\max} - R_{\max} \right) = Q_{\max}. \tag{48}$$

Thus,

$$|Q_{t+1}(s_t, a_t)| \leq Q_{\max}. \tag{49}$$

Therefore, by induction, $Q_t$ remains bounded for all $t$, independently of $\Delta_t$.

**Step 3: Verify Mean Condition**

We can write

$$\frac{1}{\gamma} \mathbb{E}[F_t] = \mathbb{E}_{p_\mathcal{T}}[G_t], \tag{50}$$

where

$$G_t = \sum_a q_t(a|s_{t+1})Q_t(s_{t+1}, a) - \max_a Q_*(s_{t+1}, a). \tag{51}$$

We can form the bound

$$\left\| \frac{1}{\gamma} \mathbb{E}[F_t] \right\|_\infty = \|\mathbb{E}[G_t]\|_\infty \leq \|G_t\|_\infty, \tag{52}$$

which means that if we can bound $\|G_t\|_\infty$ appropriately, the mean criterion will be satisfied.

Assuming that $b_t$ places $(1 - \delta_t)$ mass on the maximal action $a^* = \arg\max_a Q_t(s_{t+1}, a)$, we can write

$$G_t = \sum_a q_t(a|s_{t+1})Q_t(s_{t+1}, a) - \max_a Q_*(s_{t+1}, a) \tag{53}$$

$$= (1 - \delta_t)Q_t(s_{t+1}, a^*) + \delta_t \sum_{c \neq a^*} \tilde{q}_t(c|s_{t+1})Q_t(s_{t+1}, c) - \max_a Q_*(s_{t+1}, a), \tag{54}$$

where $\tilde{b}_t(c|s_{t+1}) = \frac{b_t(c|s_{t+1})}{\delta_t}$ for $c \neq a^*$.

We can then write

$$G_t = Q_t(s_{t+1}, a^*) - \max_a Q_*(s_{t+1}, a) + \delta_t \left( \sum_{c \neq a^*} \tilde{b}_t(c|s_{t+1})[Q_t(s_{t+1}, c) - Q_t(s_{t+1}, a^*)] \right). \tag{55}$$

Since $Q_t(s_{t+1}, a^*) \geq Q_t(s_{t+1}, c)$ for all $c$, the terms inside the brackets are non-positive. Therefore,

$$G_t \leq Q_t(s_{t+1}, a^*) - \max_a Q_*(s_{t+1}, a). \tag{56}$$

Now, we have

$$Q_t(s_{t+1}, a^*) - \max_a Q_*(s_{t+1}, a) = [Q_t(s_{t+1}, a^*) - Q_*(s_{t+1}, a^*)] + [Q_*(s_{t+1}, a^*) - \max_a Q_*(s_{t+1}, a)] \tag{57}$$

$$\leq \Delta_t(s_{t+1}, a^*). \tag{58}$$

Thus,

$$G_t \leq \Delta_t(s_{t+1}, a^*). \tag{59}$$

Therefore,

$$\|G_t\|_\infty \leq \|\Delta_t\|_\infty. \tag{60}$$

Additionally, the term involving $\delta_t$ contributes an additional $c_t$, which is bounded due to the boundedness of $Q_t$ and $\delta_t \to 0$. Thus, the mean condition becomes

$$\|\mathbb{E}[F_t]\|_\infty \leq \gamma\|\Delta_t\|_\infty + c_t, \tag{61}$$

with $c_t \to 0$ as $\delta_t \to 0$.

Since $\gamma < 1$, the mean condition is satisfied with $\kappa = \gamma$ and $c_t \to 0$.

**Step 4: Verify Variance Condition**

Since the rewards $r_t$ are bounded and we have established that $Q_t$ is bounded independently, $F_t$ is also bounded.

We can write:

$$\Delta F_t = F_t - \mathbb{E}[F_t] \tag{62}$$

$$= (r_t - \mathbb{E}[r_t|s_t, a_t]) + \gamma \left( \sum_a q_t(a|s_{t+1})Q_t(s_{t+1}, a) - \mathbb{E}_{s_{t+1}}\left[ \sum_a q_t(a|s_{t+1})Q_t(s_{t+1}, a) \right] \right). \tag{63}$$

We can bound the variance using

$$\text{Var}(F_t) = \mathbb{E}\left[ (\Delta F_t)^2 \mid \mathcal{F}_t \right] \leq \|\Delta F_t\|_\infty^2. \tag{64}$$

Using the triangle inequality,

$$\|\Delta F_t\|_\infty \leq \|\Delta r_t\|_\infty + \gamma \left\| \sum_a q_t(a|s_{t+1})Q_t(s_{t+1}, a) - \mathbb{E}_{s_{t+1}}\left[ \sum_a q_t(a|s_{t+1})Q_t(s_{t+1}, a) \right] \right\|_\infty \tag{65}$$

$$\leq \|\Delta r_t\|_\infty + \gamma \left\| Q_t(s_{t+1}, a) - \mathbb{E}_{s_{t+1}}[Q_t(s_{t+1}, a)] \right\|_\infty. \tag{66}$$

Since $Q_t$ is bounded, there exists a constant $B$ such that

$$\|Q_t(s_{t+1}, a) - \mathbb{E}_{s_{t+1}}[Q_t(s_{t+1}, a)]\|_\infty \le 2Q_{\max} = B. \tag{67}$$

Therefore,

$$\|\Delta F_t\|_\infty \le \|\Delta r_t\|_\infty + \gamma B. \tag{68}$$

Since $r_t$ is bounded, $\|\Delta r_t\|_\infty \le 2R_{\max}$.

Thus,

$$\|\Delta F_t\|_\infty \le 2R_{\max} + \gamma B. \tag{69}$$

Therefore, the variance is bounded, and there exists a constant $C > 0$ such that

$$\mathrm{Var}(F_t) \le C(1 + \|\Delta_t\|_\infty)^2. \tag{70}$$

**Step 5: Conclusion**

All the conditions of Theorem 1 are satisfied:

- **Step-Size Conditions:** Verified in Step 1.

- **Mean Condition:** Verified in Step 3, with $\kappa = \gamma < 1$ and $c_t \to 0$.

- **Variance Condition:** Verified in Step 4.

Therefore, $\Delta_t \to 0$ with probability 1, implying that $Q_t \to Q_*$ with probability 1.

$\square$

## A.4 EXPERIMENT SETTING

### A.4.1 CLASSIC CONTROL AND BOX 2D ENVIRONMENT

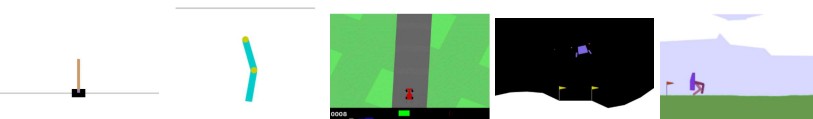

Figure 3: Cartpole, Acrobot, CarRacing, Lunar Lander and Bipedal Walker .

1. Cartpole: a pole is attached by an unactuated joint to a cart, which moves along a frictionless track. The pendulum is placed upright on the cart and the goal is to balance the pole by applying forces in the left and right direction on the cart.

2. Acrobot: a two-link pendulum system with only the second joint actuated. The task is to swing the lower link to a sufficient height in order to raise the tip of the pendulum above a target height. The environment challenges the agent's ability to apply precise control for coordinating multiple linked joints.

3. CarRacing: The easiest control task to learn from pixels - a top-down racing environment. The generated track is random in every episode.

4. Lunar Lander: It is a classic rocket trajectory optimization problem. According to Pontryagin's maximum principle, it is optimal to fire the engine at full throttle or turn off. This is why this environment has discrete actions: engine on or off.

5. Bipedal Walker: a two-legged robot attempting to walk across varied terrain. The goal is for the agent to learn how to navigate efficiently and avoid falling.

### A.4.2 METADRIVE BLOCK TYPE DESCRIPTION

Table 4: Block Types Used in Experiments

| ID | Block Type |
|----|------------|
| S | Straight |
| C | Circular |
| r | InRamp |
| R | OutRamp |
| O | Roundabout |
| X | Intersection |
| y | Merge |
| Y | Split |
| T | T-Intersection |

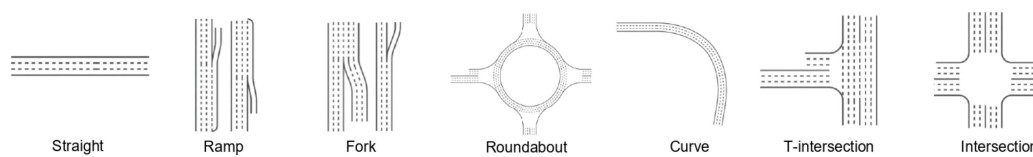

Figure 4: Various block types used in the MetaDrive environment. These blocks represent common road structures such as straight roads, ramps, forks, roundabouts, curves, T-intersections, and intersections, used for evaluating the vehicle's path planning and decision-making capabilities.

### A.4.3 MAP DESIGN AND TESTING OBJECTIVES

**Map 1: SrOYCTRyS**  This map consists of straight roads, roundabouts, intersections, T-intersections, splits, and ramps. The environment presents a highly complex combination of multiple intersections, dynamic traffic flow, and varying road structures.

**Testing Objective:** The focus of this environment is to evaluate the algorithm's smooth decision-making and multi-intersection handling, mimicking human driving behavior. The challenges include adjusting vehicle paths in real-time and ensuring smooth lane transitions in the presence of complex road structures such as roundabouts and ramps.

**Map 2: COrXSrT**  This map combines circular roads, roundabouts, straight roads, intersections, ramps, and T-intersections. The environment is designed to assess the vehicle's decision-making capabilities when dealing with continuous changes in road grades and multiple intersection types.

**Testing Objective:** This environment tests the algorithm's ability to dynamically adjust to **grade changes** and **multi-intersection interactions**, replicating human-like behavior. The goal is to observe how well the algorithm adjusts vehicle speed and direction, ensuring stability in scenarios involving ramps and complex road networks.

**Map 3: rXTSC**  This map consists of ramps, intersections, T-intersections, straight roads, and circular roads. The environment simulates multiple road interactions, testing the vehicle's path selection and stability, particularly at intersections and ramps.

**Testing Objective:** This environment evaluates the algorithm's performance in handling intersections and T-junctions with real-time path selection. The challenge is to ensure human-like adaptability when encountering multiple directional options, maintaining decision stability in dynamic traffic situations.

**Map 4: YOrSX**  This map includes splits, roundabouts, straight roads, circular roads, and intersections. The environment is tailored to test the vehicle's ability to make path decisions in high-speed settings, particularly when merging traffic and navigating through complex junctions.

**Testing Objective:** The map focuses on testing the vehicle's ability to handle **high-speed lane merging** and **dynamic path planning**. The algorithm must mimic human drivers by making real-time adjustments in a high-speed environment, choosing optimal paths while maintaining speed control and safety through complex intersections and roundabouts.

**Map 5: XTOC**  This map features circular roads, T-intersections, and straight roads, creating a unique combination of continuous curves and abrupt directional changes. The environment presents the challenge of maintaining speed while negotiating tight turns and quick transitions at T-intersections.

**Testing Objective:** The focus is on testing the vehicle's ability to handle **sharp directional changes** and maintain control during high-speed maneuvers. The algorithm needs to balance speed with precision, ensuring safe navigation through tight turns and abrupt intersections.

**Map 6: XTSC**  This map features a T-shaped intersection with traffic signals controlling vehicle flow from three directions. It tests advanced driving skills including traffic light compliance, turn management, and interaction with vehicles from cross directions.

**Testing Objective:** The main challenge is to evaluate the vehicle's ability to maintain **lane stability** and make appropriate **speed adjustments** while navigating long straight roads and transitioning into a circular roundabout. The algorithm must ensure smooth control and decision-making, simulating human-like behavior in handling both high-speed straight roads and slower, more controlled turns in the roundabout.

**Map 7: TOrXS**  This map consists of T-intersections, roundabouts, straight roads, and splits, forming a compact yet intricate structure. The layout challenges the algorithm to manage dynamic path selection and adapt to sudden directional changes within a moderately complex road network.

**Testing Objective:** The primary objective is to evaluate the algorithm's ability to manage split paths and handle sudden directional changes. The map focuses on the vehicle's adaptability in navigating roundabouts and maintaining stability while making real-time path decisions at T-intersections.

**Map 8: CYrXT**  This map integrates circular roads, Y-intersections, ramps, T-intersections, and straight roads, creating a dynamic and highly interconnected network. The layout introduces varying road geometries and frequent directional changes, requiring seamless decision-making and adaptability.

**Testing Objective:** The map is designed to test the algorithm's ability to adapt to sudden directional shifts at Y-intersections and T-junctions, maintain stability on ramps, and execute precise maneuvers on circular roads. The emphasis is on smooth transitions between road types, effective navigation through interconnected pathways, and robust handling of diverse traffic scenarios.

### A.4.4 MuJoCo Environments

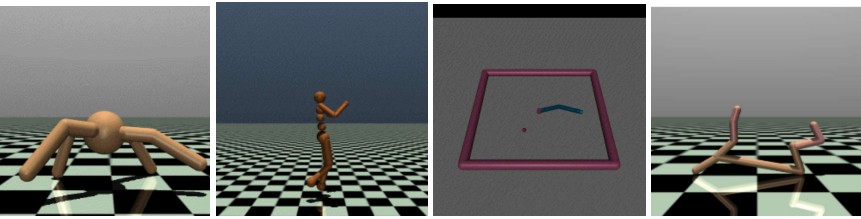

Figure 5: Ant, Humanoid, Reacher and Half Cheetah.

1. Ant: a 3D robot with a single central torso and four articulated legs is designed to navigate in the forward direction. The robot's movement depends on coordinating the torque applied to the hinges that connect the legs to the torso and the segments within each leg.

2. Humanoid: a 3D bipedal robot simulates human gait, with a torso, a pair of legs, and arms. Each leg and arm consists of two segments, representing the knees and elbows respectively; the legs are used for walking, while the arms assist with balance. The robot's goal is to walk forward as quickly as possible without falling.

3. Humanoid Standup: The environment starts with the humanoid laying on the ground, and then the goal of the environment is to make the humanoid stand up and then keep it standing by applying torques to the various hinges.

4. Reacher: a two-jointed robot arm. The goal is to move the robot's end effector close to a target that is spawned at a random position.

5. Half Cheetah: a 2-dimensional robot consisting of 9 body parts and 8 joints connecting them (including two paws). The goal is to apply torque to the joints to make the cheetah run forward (right) as fast as possible, with a positive reward based on the distance moved forward and a negative reward for moving backward.

6. Hopper: a two-dimensional one-legged figure consisting of four main body parts - the torso at the top, the thigh in the middle, the leg at the bottom, and a single foot on which the entire body rests. The goal is to make hops that move in the forward (right) direction by applying torque to the three hinges that connect the four body parts.

7. Walker-2d: a two-dimensional bipedal robot consisting of seven main body parts - a single torso at the top (with the two legs splitting after the torso), two thighs in the middle below the torso, two legs below the thighs, and two feet attached to the legs on which the entire body rests. The goal is to walk in the forward (right) direction by applying torque to the six hinges connecting the seven body parts.

8. Pusher: a multi-jointed robot arm that is very similar to a human arm. The goal is to move a target cylinder (called object) to a goal position using the robot's end effector (called fingertip).

9. Inverted Pendulum: The environment consists of a cart that can be moved linearly, with a pole attached to one end and having another end free. The cart can be pushed left or right, and the goal is to balance the pole on top of the cart by applying forces to the cart.

10. Inverted Double Pendulum: The environment involves a cart that can be moved linearly, with one pole attached to it and a second pole attached to the other end of the first pole (leaving the second pole as the only one with a free end). The cart can be pushed left or right, and the goal is to balance the second pole on top of the first pole, which is in turn on top of the cart, by applying continuous forces to the cart.

A.4.5   ATARI ENVIRONMENTS

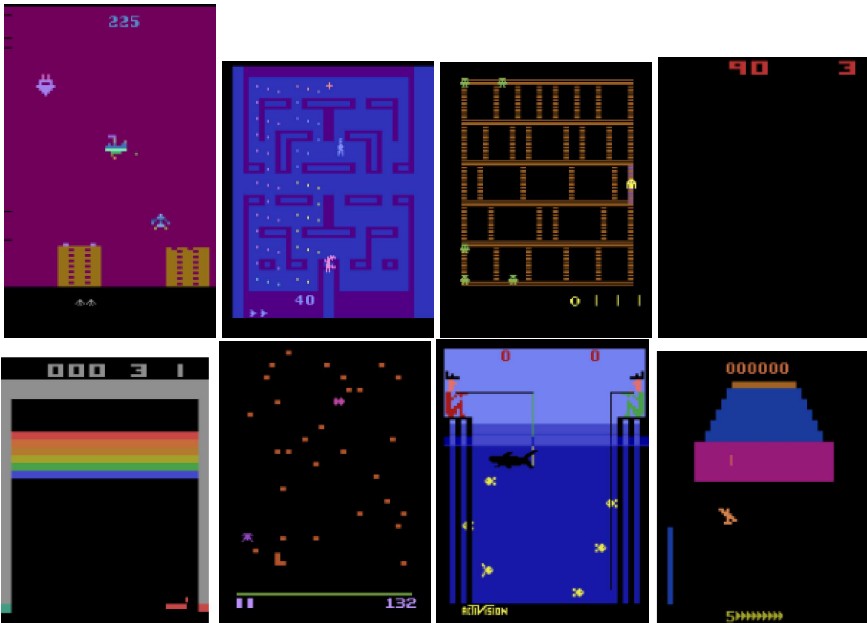

Figure 6: Air Raid, Alien, Amidar, Asteroids, Breakout, Centipede, Fishing Derby, Zaxxon.

1. Air Raid: You control a ship that can move sideways and protect two buildings (one on the right and one on the left side of the screen) from flying saucers that are trying to drop bombs on them.

2. Alien: You are stuck in a maze-like space ship with three aliens. You goal is to destroy their eggs that are scattered all over the ship while simultaneously avoiding the aliens (they are trying to kill you).

3. Admidar: You are trying to visit all places on a 2-dimensional grid while simultaneously avoiding your enemies. You can turn the tables at one point in the game: Your enemies turn into chickens and you can catch them.

4. Asteroids: You control a spaceship in an asteroid field and must break up asteroids by shooting them. Once all asteroids are destroyed, you enter a new level and new asteroids will appear. You will occasionally be attacked by a flying saucer.

5. Breakout: You move a paddle and hit the ball in a brick wall at the top of the screen. Your goal is to destroy the brick wall. You can try to break through the wall and let the ball wreak havoc on the other side, all on its own! You have five lives.

6. Centipede: You are an elf and must use your magic wands to fend off spiders, fleas and centipedes. Your goal is to protect mushrooms in an enchanted forest.

7. Fishing Derby: Your objective is to catch more sunfish than your opponent.

8. Zaxxon: Your goal is to stop the evil robot Zaxxon and its armies from enslaving the galaxy by piloting your fighter and shooting enemies.