# OpenReview forum: "Conceptual Belief-Informed Reinforcement Learning"
_ICLR.cc/2026/Conference — ICLR 2026 Conference Withdrawn Submission_

### Official Review · Reviewer_5dyR · 2025-10-28

**Soundness:** 2
**Presentation:** 1
**Contribution:** 1
**Rating:** 0
**Confidence:** 3

**Summary:**

The paper is difficult to follow as it tries to ground methodological development on cognitive science concepts such as "conceptual abstraction" and "probabilistic priors", but which are superficially discussed. Concept learning is an old ML technique (with well-known limitations) and probabilistic priors is so broad term that it needs careful definition.

What I understand (mainly from Figure 1) is that "concept" is a specific environment and its description is formed by clustering its states. It is unclear how the concept vector is generated during learning, but the "prior" is formed by summing the observed state and concept descriptor. Furthermore, it is unclear how the concept 1...K is selected during training.

**Strengths:**

I cannot judge since many important details missing or definitions are confusing

**Weaknesses:**

**Major:**

 - The idea of using "concepts" suggests that there exist previous environments agent has explored to form the concepts. This, on the other hand, means that the correct context of this work is domain adaptation or meta RL where past experiences are used to boost learning.

 - Results are reported only for the two baselines, DQN and SAC, that rather old methods and there have been many improvements for them. Even then, it is unclear in which cases the results are statistically significant as the variances are large.

 - It is unclear what environments are used to form the concepts in the experiment - and since other methods do not use them, is the comparison fair at all? The method should be compared to domain transfer and meta RL methods.

 - Writing is very difficult to follow - since the idea is rather simple, writing should be simple as well. As an example, the two contributions are so verbose that it is difficult to understand what do they really mean

**Moderate:**

 - C_k in Definition 4 are not defined. The definition S=Union of all C suggests that C_k:s are same as the states. The short description below the definition further suggests that C_k are formed by clustering the observed states to K clusters. All these should be clarified. Moreover, this approach has been used in offline RL methods.

 - Section 4.2 is unclear. Description is complex, e.g., "For given state s in S, we first identify its concept index c(s) such that s in C_c(s), and then fuse the signals" - does this simply mean that the closest cluster is identified by search and its mean vector added to the current state. It is a complicated way to say simple things.


**Minor:**

 - Try to reduce the amount of mathematical terms that can be defined using already existing ones

 - The PDF seems to be a binary dump instead of a normal text PDF. Therefore I cannot highlight any parts of the text and links do not work. Authors should instead produce a normal PDF where all these work.

**Questions:**

I think this method cannot be compared to standard RL methods, but methods that utilize past experiences to speed up training. Therefore, the context is wrong. Can you change my mind?

---

> ### Author Response · Authors · 2025-11-24
>
> We sincerely thank the reviewer for the valuable suggestions and the kind recognition of our contribution and presentation. We have also provided responses addressing the issues you raised.
> ### W1. Conceptual and Methodological Issues
> * **Unclear Use of Prior Environments and Fairness in Baseline Comparison**
>  Thank you for the insightful question. You are correct that our paper does not clearly explain how concepts are constructed. First, we clarify that our work focuses on single-environment (single-task) RL and does not involve transfer across environments (e.g., meta-RL or cross-domain RL). So there is not a process to get prior from any previous environment (or task).
>
>
> * **Lack of appropriate experiments to fairly evaluate the method**.
>     We compare our framework with more than just DQN and SAC.  We respectively hope the reviewer will evaluate our results fairly, as our experiments span DQN (value-based/discrete), PPO (on-policy actor–critic), SAC (stochastic off-policy continuous), and TD3 (deterministic off-policy continuous), covering the major paradigms in deep reinforcement learning. While those are not state-of-the-art algorithms, they are representative RL algorithms for different kinds of policies and RL tasks, which are sufficient to validate our central hypothesis: adding a concept-based belief module and plug them in SAC/DQN/PPO can significantly improve data efficiency. In future work, we plan to include more baselines of prior experience-utilization baselines, such as replay-based methods (e.g., HER [1] and prioritized variants), episodic memory models (e.g., NEC [2]), and state abstraction approaches such as bisimulation and contrastive learning.
>
>
> ### W2. Writing and Presentation Clarification
> * **Contains many mathematical formulas, making it somewhat difficult to read**
> You’re absolutely right that our paper includes many formulas, which can make it challenging to read. In the revised version, we will thoroughly reduce the mathematical detail in the main text, emphasize more intuitive explanations, and move additional derivations to the appendix.
>
> * **PDF format issues**
> The issue with the file format was an oversight on our part. We appreciate you bringing it to our attention and will definitely fix this in the new version.
>
> ### Q1. I think this method cannot be compared to standard RL methods, but methods that utilize past experiences to speed up training. Therefore, the context is wrong. Can you change my mind?
> We thank the reviewer for the thoughtful and constructive comment. We fully agree that meaningful comparisons to prior work on experience utilization are essential for evaluating our method. To address this, we plan to include additional baselines in the next version (and, if not feasible within the rebuttal window, in a subsequent revision), such as replay-based methods (e.g., HER [1] and prioritized variants), episodic memory models (e.g., NEC [2]), and state-abstraction approaches including bisimulation and contrastive learning.
>
> At the same time, we emphasize that our concept-based belief updating module is complementary to most existing experience-utilization methods—primarily data-level approaches like HER, rather than purely competitive. Integrating our concept, based belief framework can enhance data efficiency alongside these methods. Concretely, this complementarity arises because our contribution introduces a new learning rule based on abstracted concepts, not an improved data-management strategy. Methods such as PER and HER operate at the data level, controlling which transitions to replay or relabel. In contrast, HI-RL operates at the learning-rule level, introducing a Bayesian fusion mechanism that adjusts how the policy or value function is updated for each data point.
>
> > [1] Zhang, Shangtong, and Richard S. Sutton. "A deeper look at experience replay." arXiv preprint arXiv:1712.01275 (2017).
> > [2] Pritzel, Alexander, et al. "Neural episodic control." International conference on machine learning. PMLR, 2017.

---

### Official Review · Reviewer_6pJc · 2025-11-01

**Soundness:** 2
**Presentation:** 2
**Contribution:** 2
**Rating:** 2
**Confidence:** 4

**Summary:**

This paper introduces Conceptual Belief-Informed Reinforcement Learning (HI-RL), a framework that aims to improve sample efficiency in RL by organizing experiences into categories and forming adaptive probabilistic beliefs. The approach partitions states into concepts using clustering (K-means), maintains a belief distribution over actions for each concept based on historical experience, and combines these beliefs with the learning target in standard RL algorithms (DQN, PPO, SAC, TD3). The authors evaluate HI-RL across discrete and continuous control tasks, demonstrating improvements in sample efficiency and final performance compared to baseline methods.

**Strengths:**

1. The proposed method is algorithm-agnostic. The authors demonstrate its ability to integrate with multiple RL algorithms (Q-learning, PPO, SAC) demonstrates versatility.
2. The mathematical formulation is generally correct and the integration mechanism is well-defined. I appreciate the derivation of convergence guarantee on the proposed smoothed Bellman Operator in the appendix.
3. Experimental evaluation spans multiple domains and algorithms, showing consistency.
4. I appreciate the good motivation from cognitive science literature to solve an important RL challenge (sample efficiency).

**Weaknesses:**

1. As the main claim of the submission is to propose a new “experience utilization paradigm,” the critical comparison baselines should be comparing to other methods aimed at improving experience utilization, including replay methods (e.g. HER and its prioritized variants), episodic memory models (NEC), and state abstraction methods such as bi-simulation and contrastive learning (Patil et al, 2024). The current comparison baselines use only the base RL algorithms, which do not seem to be reasonable and interesting comparisons. This is perhaps the major weakness of this submission.
2. No ablation studies/ sensitivity analysis were performed. Unclear what the critical hyper parameter values were (the limiting weight beta^*, the number of categories k), and how their presence/ value affect the results. The concept formation definition (Definition 4.1) requires predetermined k but provides no guidance on selection.
3. Since the concept formation is trained using k-means clustering, it is unclear whether the clusters become “semantically coherent category”, as the authors claimed. If the k-means are done on the pixel space, it seems unlikely a simple k-means clustering will render task-relevant, semantically-meaningful categories. (Please provide more evidence on this if any.) Given this, I’d be hesitant to call the proposed algorithm “concept formation.” Also, no discussion on how concept quality affects downstream performance.
4. Experiment results reporting should be better presented. Some improvements could be within noise (e.g. on CartPole, 499.78+/-0.22 seems not significantly different from 499.17+/-0.83). If proper statistical tests were done, please report the results. Otherwise use bold for all comparable performance
5. The naming of adaptive belief is somewhat misleading/ confusing. As belief is more commonly used in RL within POMDP context to refer to the posterior of environment states given all the history; in contrast, in this submission, what the authors called belief is actually “the integrated action preferences of all states within the same category.” It’d be better to use other terms for this marginal distribution to prevent confusion.

**Questions:**

1. Can you provide visualizations of the learned concepts? Do they correspond to meaningful task structure/ semantics (e.g., in Atari, do concepts capture semantic game states)?
2. Have you experimented with other clustering methods or learned representations? Why is K-means sufficient?
3. The fusion mechanism seems to be ad-hoc weighted averaging. Can you provide an explanation on why this is designed in this way? β scheduling (β* with β^t) seems to introduce additional hyperparameters without principled justification.
4. What is the effect of different β scheduling schemes?
5. What is the computational overhead compared to baselines? How does this scale with buffer size and state dimensionality?
6. In what scenarios does HI-RL perform worse than baselines? Can you characterize when conceptual priors mislead learning?
7. Unsure if this only happens to the files I downloaded, but the pdf was image based (not able to select or highlight text), making it hard to review the submission materials. Please upload a functional file is possible.

---

> ### Author Response · Authors · 2025-11-24
>
> We sincerely thank the reviewer for the suggestions and recognition of our work.We sincerely thank the reviewer for the constructive feedback and thorough evaluation of our work. We have also provided responses addressing the issues you raised.
> ### W1. Insufficient cross-method comparisons have been conducted to rigorously assess the framework’s performance
> * **Comparison Baselines (Experience Utilization)**
> We acknowledge that our current baselines include only standard RL algorithms. In future work, we plan to compare HI-RL against prior experience-utilization baselinesm, such as replay-based methods (e.g., HER [1] and prioritized variants), episodic memory models (e.g., NEC [2]), and state abstraction approaches such as bisimulation and contrastive learning. Our present experiments are designed primarily to demonstrate the framework’s algorithm-agnostic, modular design and its seamless integration across multiple standard RL algorithms. We also emphasize that our module can be integrated with existing experience-utilization baselines, making the relationship not merely comparative but complementary: integrating our concept-based belief framework can provide mutual benefits.
> > [1]Zhang, Shangtong, and Richard S. Sutton. "A deeper look at experience replay." arXiv preprint arXiv:1712.01275 (2017).
> > [2]Pritzel, Alexander, et al. "Neural episodic control." International conference on machine learning. PMLR, 2017.
>
> * **Ablation and Hyperparameter Sensitivity**
> We agree that systematic ablation and sensitivity analyses are important. Future studies will rigorously examine how the limiting weight $\beta$ and its scheduling strategy influence both concept formation and policy performance. Building on our ablation findings, we will further analyze why smaller maximum $\beta$ leads to more stable and consistent learning, and systematically evaluate alternative scheduling schemes beyond the full-trajectory and fixed-value settings. [1]
>
> In addition, our added ablation study of  the number of clusters $k$ shows that, the performance is not very sensitive to $k$ if they are in a reasonable range (not too large or too small). A overly large $k$ may lead to an excessively fine partitioning of the state space, resulting in insufficiently informative briefs. Conversely, when $k$ is too small, each cluster contains too many state samples, which in turn degrades the quality of the resulting brief information. [1]
> >[1]https://ycn6jbifyo3n.feishu.cn/wiki/RX47w49T3iDsXxkV6cycQOn9nq7?from=from_copylink
>
> * **Concept Formation Quality**
> We appreciate the reviewer’s question and acknowledge that k-means clustering, especially in pixel space, may not always yield semantically coherent concept abstractions. Our primary evidence for the meaningfulness of the abstraction is the substantial data efficiency achieved when learning with these concepts, though we recognize this is somewhat indirect. To strengthen this claim, we plan to provide additional evidence of alignment between the learned concepts and task-relevant behaviors, and to explore richer state representations, feature embeddings, and alternative clustering methods to improve semantic coherence.
>
>
> ### W2. Shortcomings in the clarity of writing and the presentation of experimental results
> * **Low experimental Reporting and Statistical Significance**
> We thank the reviewer for pointing this out and fully agree. As suggested, in the revised version we will highlight superior performance scores based on both their mean and their mean plus standard deviation.
>
> * **Terminology of Adaptive Belief**
> We recognize that the term “belief” can be misleading, as it may refer to different notions (e.g., a POMDP posterior or a cognitive-science construct). In HI-RL, it specifically denotes action preferences conditioned on each constructed concept. To improve clarity, we will adopt more precise terminology—such as “concept-based action belief”—in the revised version.
>
> ### Q1. Can you provide visualizations of the learned concepts? Do they correspond to meaningful task structure/ semantics (e.g., in Atari, do concepts capture semantic game states)?
>
> Yes! We include a 3D projection visualizations [1]of the learned concepts in Cart Pole environment with 150,000 runing steps to generate state samples， where states belonging to the same category are colored identically, clearly illustrating category distinctions and the overall partitioning.
>
> >[1]https://ycn6jbifyo3n.feishu.cn/wiki/DE26wJ9O7ilRb8kVMNHcqTFyn4f?from=from_copylink

---

> ### Author Response · Authors · 2025-11-24
>
> ### Q2. Have you experimented with other clustering methods or learned representations? Why is K-means sufficient?
> We have not yet extensively explored alternative clustering algorithms. In our framework, we adopt k-means primarily because it is widely applicable, easy to implement [1], and more computationally efficient than many alternatives. Moreover, k-means is sufficient to demonstrate superior efficiency across the control and robotics tasks studied in this work. The proposed HI-RL supports flexible selection of any concept-abstraction method, which is then combined with our Bayesian concept-based belief update and integrated with any existing RL algorithm. We plan to accommodates other advanced representations (e.g., contrastive or bisimulation-based) to gain further efficiency and concept coherance in the future work.
>
> > [1] Vora, Pritesh, and Bhavesh Oza. "A survey on k-mean clustering and particle swarm optimization." International Journal of Science and Modern Engineering 1.3 (2013): 24-26.
> ### Q3. The fusion mechanism seems to be ad-hoc weighted averaging. Can you provide an explanation on why this is designed in this way? β scheduling (β* with β^t) seems to introduce additional hyperparameters without principled justification.
> Thanks so much for this insightful quesion and here is the principle we follow: the design of β can be viewed as a simplified implementation of the Bayesian principle of “combining prior knowledge with new evidence to guide decisions”[1-3].   The scheduling of β simulates the natural transition in human learning from relying on prior experience to relying on current feedback. Within the Bayesian framework, the posterior can be approximately regarded as a weighted combination of the prior and new observations; in our method,
>
>  * $b_t$ is the prior formed based on past conceptual experience,
>  * $Z_t$ is the likelihood derived from the current state information,
>  * $B_t$ is the posterior decision preference resulting from the integration of the two.
>
> The scheduling of β is intuitive: early in training, when the model is unstable and estimates are noisy, a higher weight on the prior $𝑏_𝑡$ stabilizes learning; as experience grows and $𝑍_𝑡$ becomes more reliable, the weight shifts toward current state information. This mirrors the human transition from relying on past experience to relying on immediate judgment. Maintaining 𝛽∗ >0 preserves a minimal prior influence, mitigating overfitting to recent observations. Our convergence analysis in Appendix 3 provided the prior weight remains moderate, which is guaranteed to converge.
> > [1] Fernández, Fernando, and Manuela Veloso. "Probabilistic policy reuse in a reinforcement learning agent." Proceedings of the fifth international joint conference on Autonomous agents and multiagent systems. 2006.
> > [2] Fernández, Fernando, Javier García, and Manuela Veloso. "Probabilistic policy reuse for inter-task transfer learning." Robotics and Autonomous Systems 58.7 (2010): 866-871.
> > [3] Murphy, Kevin P. Machine learning: a probabilistic perspective. MIT press, 2012.
>
>
> ### Q4. What is the effect of different β scheduling schemes?
> Different $\beta$ values produce noticeably different training dynamics and state behaviors. To analyze its impact more precisely, we conduct two types of ablations based on our dynamic scheduling strategy. First, we vary the maximum $\beta$ while keeping all other settings fixed, and find that smaller $\beta$ consistently yields better performance with lower variance. Second, we vary the schedule, comparing full-trajectory, half-trajectory, and fixed-value (0.1) settings. The schedule has only minor influence, though using full-trajectory results in slightly more stable learning.[1]
> >[1] https://ycn6jbifyo3n.feishu.cn/wiki/RX47w49T3iDsXxkV6cycQOn9nq7
> ### Q5. What is the computational overhead compared to baselines? How does this scale with buffer size and state dimensionality?
> Thank you to the reviewer for this insightful question. The computation cost is algorithm-dependent. For example, in AirRaid, when comparing PPO and HI-PPO under the same parameter settings, the per-iteration runtime is approximately 1.0 vs. 1.2, averaged over 50M steps—about a 20% increase. By contrast, the impact of buffer size on overall computation cost is relatively minor.

---

> ### Author Response · Authors · 2025-11-24
>
> ### Q6. In what scenarios does HI-RL perform worse than baselines? Can you characterize when conceptual priors mislead learning?
>
> HI-PPO does encounter certain challenges in some environments. In particular, we observed that in several Atari environments, such as Tennis and Battle Zone, where the state is image, representing a relatively high-dimensional setting. We suspect that during concept category initialization with high-dimensional image states, compressing them into a 1D representation may lead to substantial information loss. This can result in suboptimal k-means concept abstraction, misaligned stored summaries, and ultimately hinder training, thereby limiting effective learning.
> ### Q7. Unsure if this only happens to the files I downloaded, but the pdf was image based (not able to select or highlight text), making it hard to review the submission materials. Please upload a functional file is possible.
> The issue with the file format was an oversight on our part. We appreciate you bringing it to our attention and will definitely fix this in the new version.

---

### Official Review · Reviewer_kNCF · 2025-11-01

**Soundness:** 2
**Presentation:** 3
**Contribution:** 4
**Rating:** 6
**Confidence:** 4

**Summary:**

HI-RL proposes to enhance RL by introducing conceptual abstractions (state clustering via K-means) and belief-informed updates (combining per-cluster priors with the current policy/Q-values).
In practice, the method defines clusters (“concepts”) in the state space, aggregates action statistics within each, and then fuses those distributions with the agent’s instantaneous estimates using a weighting factor.
This structure is applied to several base algorithms (DQN, PPO, SAC, TD3). The empirical results are surprisingly strong, showing consistent gains across multiple benchmarks despite the conceptual simplicity.

**Strengths:**

- **Simple yet effective regularization:** The belief blending smooths updates, reduces variance, and improves sample efficiency.
- **Experience reuse:** Clustering allows shared learning across similar states, enhancing generalization.
- **Stability and transfer:** Beliefs act as priors, providing memory and consistency across episodes.
- **Broad applicability:** Can be plugged into existing algorithms (DQN, PPO, SAC, TD3) without major architectural changes.
- **Empirically strong:** Demonstrates substantial performance improvements over standard baselines, suggesting practical value despite theoretical simplicity.

**Weaknesses:**

- **Conceptually shallow:** Essentially applies K-means for context extraction and a Bayesian-style weighted update; not a fundamentally new algorithm.
- **Lack of fair baselines:** Compared to plain SAC/PPO/TD3 but not to other context-aware approaches.
- **Overuse of cognitive/“neuroscience” framing:** Adds narrative flair but little real mechanism.
- **Concept formation quality:** If clustering fails to align with true behavioral modes, the priors can mislead learning.
- **Hyperparameter sensitivity:** Performance depends on the number of clusters (K), β-schedule, and distance metric.
- **Scalability:** K-means and per-concept beliefs may not scale well to large, high-dimensional environments.
- **Non-stationarity:** Fixed clusters may become outdated as the policy distribution shifts.
- **Computational overhead:** Maintaining and updating beliefs adds additional bookkeeping.

**Questions:**

1. How can the concepts evolve online rather than staying fixed after initial clustering?
2. How does this approach interact with **latent-space representation learning**. Could clustering be done in learned embeddings?
3. Can the concept-belief mechanism **transfer** across tasks or domains, not just within one environment?
4. What happens if concept boundaries are learned end-to-end instead of using K-means?
5. Would combining this with **contrastive encoders** yield more meaningful concepts?

---

> ### Author Response · Authors · 2025-11-24
>
> We sincerely thank the reviewer for the constructive feedback and the thoughtful assessment of our work. We sincerely thank the reviewer for the constructive feedback and thorough evaluation of our work. We have also provided responses addressing the issues you raised.
>
> ### W1. Clarification of technical novelty
>  * **Essentially just K-means plus a weighted update, with limited novelty**
>   We appreciate the reviewer’s observation. We emphasize that the proposed HI-RL is a general, algorithm-agnostic, modular framework for efficient experience reuse. It supports flexible selection of any concept-abstraction method, which is then combined with our Bayesian concept-based belief update and integrated with any existing RL algorithm. K-means is merely one illustrative example of a concept-abstraction approach used in this work; the proposed framework readily accommodates other advanced representations (e.g., contrastive or bisimulation-based).
>
>  *  **Overuse of cognitive/“neuroscience” framing**
>   We thank the reviewer for the feedback and will be more mindful and cautious when referencing these concepts. The use of cognitive terminology is mainly to convey design intuition, drawn from established findings in cognitive science, that efficient learning benefits from forming abstractions and updating probabilistic beliefs. HI-RL uses this perspective to motivate the transition from raw replay to structured experience organization, with “concept” (state grouping) and “belief” (adaptive prior) providing an interpretable vocabulary. More than the concept inspiration, the contribution of the framework is primarily algorithmic: concepts are computed (e.g., via clustering), beliefs are formally defined and integrated with policy updates, and the observed performance improvements arise from this mechanistic integration rather than from the cognitive analogy, which primarily serves to clarify the intuition.
> ### W2. Concerns about Modeling robustness and experimental adequacy
>  * **Lack of Fair Baselines**
> We agree that comparing only against SAC, PPO, or TD3 is indeed somewhat limited. Our current experiments are primarily intended to demonstrate the feasibility of the proposed framework. In future work, we plan to broaden our evaluation to include other experience-utilization baselines such as HER[1], as well as alternative concept-abstraction methods such as VQ-VAE[2], thereby providing a more comprehensive assessment of HI-RL’s relative performance.
>
>
> > [1]Zhang, Shangtong, and Richard S. Sutton. "A deeper look at experience replay." arXiv preprint arXiv:1712.01275 (2017).
> > [2]Pritzel, Alexander, et al. "Neural episodic control." International conference on machine learning. PMLR, 2017.
>
>  * **Potential Misalignment in Clustering**
> We acknowledge that if clustering does not accurately capture true behavioral modes, the resulting priors could misguide learning. Refer to the ablation study first, since that kinds of showing that performance won't be too sensitive to clustering to some extent. In the future work, as the reviewer suggested, we plan to investigate adaptive clustering techniques, alternative distance metrics, and enhanced state-processing strategies to improve alignment between learned concepts and actual behavioral patterns.
>
>  * **Hyperparameter sensitivity, such as β**
> HI-RL performance can be sensitive to hyperparameters such as the number of clusters K, the β schedule, and distance metrics for clustering. In future work, we intend to conduct systematic hyperparameter studies and ablation experiments to better understand and address these sensitivities.
>
> * **Fixed clusters may become outdated as the policy distribution shifts**
>  The reviewer's intuition is very insightful. We recognize that fixed clusters may become less representative as the policy distribution evolves. Currrently, to mitigate this, the beliefs associated with each concept are continuously updated, although the clustering itself is static within a task. So in the future, as the reviewer suggestd, we plan to investigate adaptive clustering techniques for adaptive concept self-evolving.

---

> > ### Author Response · Authors · 2025-11-24
> >
> > ### W3.Scalability and computational overhead
> >  * **Scalability of K-means and per-concept beliefs in large or high-dimensional environments**
> > Thank you for the reviewer’s question. To address the scalability of K-means, we plan to explore more scalable clustering methods as alternatives for concept abstraction. Regarding per-concept beliefs, concepts typically serve as low-dimensional representations of high-dimensional environments, so their number is usually manageable. Even if the concept count grows, we can employ dimensionality reduction and memory-efficient belief representations to maintain scalable belief updates for complex tasks.
> >
> > * **Updating beliefs increases computational overhead**
> >     We acknowledge the added cost of maintaining belief states. The incremental complexity of HI-Q is an additive
> > O(TKd) per training run, where T is the horizon, K the number of concepts, and d the state dimension. This is on top of the base algorithm’s original complexity O(T(A+M)), where 𝐴 denotes the cost of action selection and 𝑀 the cost of neural network updates. This overhead scales linearly with the base algorithm and remains small in practice because K is fixed and low. Empirically, the resulting gains in sample efficiency more than compensate for the added computation. Future work will explore more efficient update rules, sparse belief parameterizations, and batched fusion to further reduce overhead without sacrificing the advantages of concept-driven belief integration.
> > ### Q1. How can the concepts evolve online rather than staying fixed after initial clustering?
> >
> > It is a very promosing direction to do concept evolving rather than fixing concepts remain fixed. In the future, we plan to propose one/several ways to do concept evolving such as state abstraction and VQ-VAE [1] as future work.
> >
> > > [1] Van Den Oord, Aaron, and Oriol Vinyals. "Neural discrete representation learning." Advances in neural information processing systems 30 (2017).
> >
> > ### Q2. How does this approach interact with latent-space representation learning. Could clustering be done in learned embeddings?
> >
> > This is definitely possible and is indeed the advantages of our proposed concept-based belief framework. Advancated methods like latent-space representation learning or state abstraction could further enhance concept abstraction, especially for high-dimensional inputs. We use k-means for its broad applicability, simplicity, and computational efficiency, as an example to demonstrates the data efficiency of our framework [1].
> > > [1] Vora, Pritesh, and Bhavesh Oza. "A survey on k-mean clustering and particle swarm optimization." International Journal of Science and Modern Engineering 1.3 (2013): 24-26.
> >
> > ### Q3. Can the concept-belief mechanism transfer across tasks or domains, not just within one environment?
> > Yes! Our original intention was to design a plug-and-play framework that can be used in widespread settings. Although our current experiments focus on a single task/environment, the cognitive–belief fusion mechanism can be applied on top of widely used RL algorithms without altering their implementations. Consequently, the concept–belief mechanism is well-suited for transfer learning and meta-learning across domains—for example, by adding the concept and belief update modules to existing cross-task [1] and meta-RL [2] algorithms. Moreover, our concept–belief updating module may offer even greater benefits for generalization in cross-domain settings than in a single environment, since high-level concepts can often be shared across diverse tasks.
> > >[1] Zhang, Amy, Harsh Satija, and Joelle Pineau. "Decoupling dynamics and reward for transfer learning." arXiv preprint arXiv:1804.10689 (2018).
> > >[2] Melo, Luckeciano C. "Transformers are meta-reinforcement learners." international conference on machine learning. PMLR, 2022.
> >
> > ### Q4 & Q5 Can end-to-end learning of concept boundaries with contrastive encoders yield more meaningful concepts than K-means?
> > The reviewer is correct that replacing K-means with more advanced end-to-end concept abstraction methods,such as VQ-VAE[1], could yield higher-quality state-based concept abstractions. In addition, the suggestion to employ contrastive encoders is excellent, as it can enrich concept expressiveness by increasing inter-concept separation. Indeed, enhancing concept processing—whether via more expressive feature extraction from raw states or more advanced clustering—represents a promising direction for future work. At the same time, these alternatives typically incur higher computational costs, and we will remain mindful of this trade-off.
> > > [1] Van Den Oord, Aaron, and Oriol Vinyals. "Neural discrete representation learning." Advances in neural information processing systems 30 (2017).

---

### Note · Authors · 2025-12-01

**Comment:**

The following link [1] shows our simple HER baseline used as an ablation in a single-task setting. However, single-task environments are generally not well suited for applying HER; it is typically more meaningful in complex, multi-task–specific scenarios. Nevertheless, this does not mean that our comparative experiments are invalid.

Within single-task domains, our HI-RL framework can be applied to any method that relies on the Bellman operator, and this learning approach can still lead to potential improvements. This demonstrates that our method does not impose constraints on task settings or other assumptions.

More importantly, HI-RL aims to introduce a fundamentally new way of learning in reinforcement learning. Our goal aligns with the major challenge highlighted by Ilya [2]: value functions struggle to achieve human-like generalization. HI-RL is designed to explore this challenge directly.
>[1]https://ycn6jbifyo3n.feishu.cn/wiki/FE9TwEfqPiZLqMk2NtycDhYGn1g?from=from_copylink

>[2]https://www.youtube.com/watch?v=aR20FWCCjAs

**Withdrawal Confirmation:**

I have read and agree with the venue's withdrawal policy on behalf of myself and my co-authors.